# THE RAMANUJAN LIBRARY - AUTOMATED DISCOVERY ON THE HYPERGRAPH OF INTEGER RELATIONS

**Itay Beit-Halachmi and Ido Kaminer**
The Ramanujan Machine Team, Faculty of Electrical and Computer Engineering
Technion - Israel Institute of Technology
Haifa 3200003, Israel
`itaybe@campus.technion.ac.il, ido.kaminer@gmail.com`

## ABSTRACT

Fundamental mathematical constants appear in nearly every field of science, from physics to biology. Formulas that connect different constants often bring great insight by hinting at connections between previously disparate fields. Discoveries of such relations, however, have remained scarce events, relying on sporadic strokes of creativity by human mathematicians. Recent developments of algorithms for automated conjecture generation have accelerated the discovery of formulas for specific constants. Yet, the discovery of connections between constants has not been addressed. In this paper, we present the first library dedicated to mathematical constants and their interrelations. This library can serve as a central repository of knowledge for scientists from different areas, and as a collaborative platform for development of new algorithms. The library is based on a new representation that we propose for organizing the formulas of mathematical constants: a hypergraph, with each node representing a constant and each edge representing a formula. Using this representation, we propose and demonstrate a systematic approach for automatically enriching this library using PSLQ, an integer relation algorithm based on QR decomposition and lattice construction. During its development and testing, our strategy led to the discovery of 75 previously unknown connections between constants, including a new formula for the 'first continued fraction' constant $C_1$, novel formulas for natural logarithms, and new formulas connecting $\pi$ and $e$. The latter formulas generalize a century-old relation between $\pi$ and $e$ by Ramanujan, which until now was considered a singular formula and is now found to be part of a broader mathematical structure. The code supporting this library is a public, open-source API that can serve researchers in experimental mathematics and other fields of science.

## 1 INTRODUCTION

With the rise in computing power and the rise of artificial intelligence, efforts have been made to harness these tools to further scientific discovery. Some of the earliest examples of such automated discovery efforts include the Automated Mathematician and later Eurisko (Lenat and Brown, 1984), systems that automatically discovered concepts in mathematics and later in other scientific domains. Another notable early discovery system is Graffiti (Fajtlowicz, 1988), which automatically generated conjectures in graph theory. Since then, the usage of computer algorithms as scientific tools (Wang, 1960; Davis and Lenat, 1982; Wolfram et al., 2002), and in particular as tools for aiding with mathematical proofs (Appel and Haken, 1976; Zeilberger, 1990; Wilf and Zeilberger, 1992; Buchberger et al., 2006), has skyrocketed. Noteworthy recent examples include papers by Google DeepMind (Davies et al., 2021; Fawzi et al., 2022; Romera-Paredes et al., 2023), which have utilized AI to find new algorithms for matrix multiplication, and generate conjectures in topology, representation theory, and combinatorics.

A noteworthy example of computers assisting mathematicians is the Ramanujan Machine project (Raayoni et al., 2021; Razon et al., 2023; Elimelech et al., 2024), which specializes in automated conjecture generation in number theory. This project, named in honor of Srinivasa Ramanujan's substantial mathematical contributions (Berndt, 2012) and his unconventional work style, demonstrated

an automated discovery of formulas involving fundamental mathematical constants. The discovered formulas and the properties they exhibited (Ben David et al., 2024) enabled the construction of a previously unknown mathematical structure (David, 2023; Elimelech et al., 2024) that forms a systematic framework for the generation of new formulas and for proofs of irrationality of constants, even in places where no systematic proofs were known (e.g., Apéry (1979); David (2023)).

Fundamental mathematical constants like $\pi$, $e$, or $\zeta(3)$ are central in several branches of mathematics and in other fields of science (Finch, 2003). The appearance of the same mathematical constant in different fields that may seem unrelated can lead to surprising connections. One of the earliest such connections is the Basel problem, posed in 1650 and solved by Euler (Euler, 1748; Ayoub, 1974), establishing that one of the values of the Riemann zeta function is $\zeta(2) = \pi^2/6$, and also providing a family of formulas expressing all positive even values $\zeta(2n)$ using $\pi^{2n}$. Due to the connection of the Riemann zeta function to the distribution of prime numbers, this formula established the profound and nontrivial importance of $\pi$ in regards to prime numbers. This example shows how discovery of new connections on fundamental constants has great potential in connecting different fields and leading to new discoveries. Another such example is a connection involving $\pi$ and $e$ found by Ramanujan, presented in figure 1. Though it is not known if $\pi$ and $e$ are algebraically independent Morandi (1996), this is one of many formulas that show how else they can be connected.

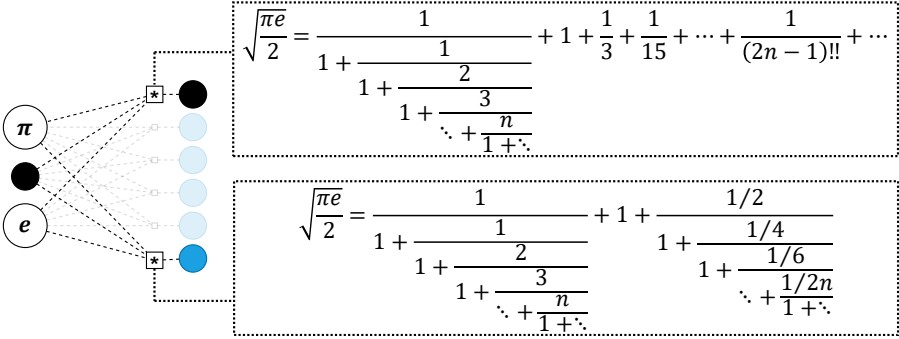

Figure 1: **Automatically discovered formulas**, showing a section of the full hypergraph in figure 4. The first formula was found by Ramanujan (Berndt et al., 1999), and the second formula is novel.

In this paper, we propose a new approach to the discovery of relations between mathematical constants, and collect its results in a publicly-accessible database. This database is the first to collect and organize the known mathematical constants and their relations, and will provide a unique resource for experimental mathematicians and number theorists worldwide.

We expand our database into a library that also collects the known relations between the constants and organizes them into a hypergraph, which we show to be an effective representation of the structure among formulas of mathematical constants and among the constants themselves. Going beyond the collection of known relations, we develop algorithms that use the database to search for new relations between its constants, successfully generating 75 new formulas. These algorithms are used for further (automated) enrichment of the library. We present two enrichment strategies: (1) automated search of relations within the database using a polynomial integer relation algorithm (see section 2.1) and (2) the *identify* algorithm (see section 4) executed on each new constant introduced to the library to find its relations to existing constants and formulas in the database. Both strategies rely on the integer relation algorithm PSLQ (Ferguson and Bailey, 1992; Ferguson et al., 1999). We developed a new methodology for quantifying the PSLQ results and for determining which ones constitute new likely formulas, so they can be further analyzed to greater precision. Our algorithm is the first to discover nonlinear (polynomial) relations, rather than focusing on linear relations. For comparison, we show that *identify* succeeded in cases for which the state-of-the-art commercial solution (in Wolfram Alpha) was unsuccessful.

## 2 INTRODUCING THE HYPERGRAPH OF MATHEMATICAL CONSTANTS AND THEIR RELATIONS

Recall that an (ordinary) undirected *graph* on a *vertex* set $V$ is defined by an *edge* set $E$, each connecting two vertices. An undirected *hypergraph*, then, generalizes the concept of an edge so each one can connect more than two vertices. More explicitly, each edge is now a set of arbitrary size, instead of a set of size 2. In our (undirected) hypergraph, the vertices will all be mathematical constants, either given explicitly (e.g., $\pi, e$, up to a user-defined precision) or given by a formula converging to the constant. We choose to represent formulas using a canonical form that captures general continued fractions and infinite sums, namely the $\mathcal{C}$-transforms of a complex sequence $f_n$:

$$\mathcal{C}[f_n] = 1 + \cfrac{f_1}{1 + \cfrac{f_2}{1 + \cfrac{f_3}{\ddots}}}. \tag{1}$$

This definition can always be treated as a formal expression, and when it converges it is instead defined as the limiting value. In practice, we generate each $f_n$ using a rational function. $\mathcal{C}$-transforms (and continued fractions in general) are powerful as they contain, for example, all infinite sums (that can be converted using Euler's approach (Euler, 1748)), whereas not every continued fraction can be converted back to an infinite sum. The $\mathcal{C}$-transform captures any continued fraction in a canonical form that removes redundancy. The special case of $f_n$ being a rational function already generalizes all polynomial continued fractions (Khinchin, 1964) , which are general mathematical constructs that capture all trigonometric functions, Bessel functions, generalized hypergeometric functions and much more. The $\mathcal{C}$-transform notation helps facilitating the systematic automated research in our work (see appendix A for further discussion).

The undirected edges (equivalently hyperedges) in our hypergraph are *integer polynomial relations*, forming a representation that generalizes a large number of formula structures. To define an integer polynomial relation, we first introduce integer relations: An integer relation on a vector of real numbers $x_1, x_2, ...$ is a vector of integers $a_1, a_2, ...$, not all of which are 0, such that $a_1x_1 + a_2x_2 + ... = 0$. As an example, the formula $\phi = \frac{\sqrt{5}+1}{2}$ is equivalent to the integer relation $2\phi - \sqrt{5} - 1 = 0$, with the constants being $\phi, \sqrt{5}, 1$ and the integer coefficients being $2, -1, -1$. This example illustrates how integer relations are a general structure that captures mathematical equalities.

We expand on this definition by introducing *polynomial relations*, which have $a_1, a_2, ...$ as the integer coefficients of a (nonzero) multi-variable polynomial $p$, looking for a solution satisfying $p(x_1, x_2, ...) = 0$. For each identified polynomial, we define its *degree* as the greatest sum of all exponents in each monomial, and its *order* as the largest exponent that appears in it. These quantities let us measure how "complex" a given polynomial is, which we use for eliminating redundancies in our algorithm. As another example, consider the relation between $\pi$ and $e$, captured by the Ramanujan formula in figure 1. These types of relations constitute the edges of the hypergraph we construct automatically in this work.

To account for possible numerical inaccuracies arising from our computational methods, we propose a definition that allows for a (typically small) error $\varepsilon := |a_1x_1 + a_2x_2 + ...| \geq 0$. This definition also applies to polynomial relations, in which case $\varepsilon = |p(x_1, x_2, ...)|$. Then, the *precision* of the relation is defined as $\lfloor -\log_2 \varepsilon \rfloor$. With the ability to quantify each constant's numerical error as $\varepsilon_1, \varepsilon_2, ...$, we instead replace $\varepsilon$ in the precision with $\max\{\varepsilon, \varepsilon_1, \varepsilon_2, ...\}$, allowing the discovered relation to account for the inaccuracy of its constants.

Hypergraph edges stand for either linear relations with integer coefficients, like that between $\sqrt{5}$ and $\phi = \frac{\sqrt{5}+1}{2}$, or nonlinear (polynomial) relations, like that between $\pi$ and $\zeta(2) = \frac{\pi^2}{6}$. Interestingly, all edges representing linear relations satisfy a kind of transitivity in any subgraph they form: given an integer relation on a set of constants $A$, and another on a set of constants $B$, such that $A$ and $B$ share at least one constant $x$, it is possible to construct a relation on $(A \cup B) \setminus \{x\}$. However, edges like the ones we show between $\pi$ and $e$ impose a more intricate structure that does not conform to regular transitivity but requires its generalization. A simple example of such transitivity is the (non-linear) polynomial relations between $\pi$ and its higher powers, or $e$ and its rational exponents.

## 2.1 ALGORITHMS FOR AUTOMATED ENRICHMENT OF THE HYPERGRAPH

Our algorithms populate the hypergraph of integer relations by searching for sets of constants $x_1, x_2, ...$ and polynomials $p$ such that $p(x_1, x_2, ...) = 0$ up to some tolerance. Section 3.1 summarizes the results, with a full listing in appendix F. The underlying mechanism is as follows:

Beginning with a totally disconnected hypergraph (i.e. no relations are known yet), choose the search space of constants, partitioned into user-defined subsets, one of which may be fundamental mathematical constants, and another may be constants provided as limits of formulas (generally described by $\mathcal{C}$-transforms). Then, each run of the algorithm takes some or all of the subsets in the partition, filters them further, and then takes their product space. In addition, the algorithm requires a choice of maximum degree and order for the polynomials. For example, the hypergraph in figure 1 resulted from a search that yielded an edge of degree $d = 6$ and order $o = 2$ for the polynomials on $\pi$ and $e$, which can be represented using a simpler degree 5 edge on the constant $\pi e$ instead.

Second, once the search space is decided, each set of constants is run through an integer relation algorithm to obtain a polynomial relation, using multiple precision arithmetic (see section 3 for details). For this we have chosen the PSLQ algorithm, whose supporting theorems ensure that it can recover a relation with the working precision for almost all real vectors (Ferguson et al., 1999), except for some special cases we note in appendix B. We discuss in the next section how we run PSLQ and what heuristic we use to filter integer relations that are likely to be false positives. This is the most resource-intensive part of the algorithm, with each run of PSLQ having a time complexity of $O(m^4)$ where $m$ is the maximal number of monomials in a polynomial with degree $d$ and order $o$ (see appendix C for a more detailed analysis). However, each run of PSLQ is independent from every other run, making the algorithm embarrassingly parallel, allowing us to scale the search space with the number of processors available.

Lastly, once significant relations have been collected in the previous step to populate the hypergraph, these integer relations are then used in future runs of the algorithm to save time: Given a set of constants $X$, if the hypergraph already has an edge $e \subseteq X$ whose degree and order are no more than the current maximal degree and order, then we do not run $X$ through PSLQ again. Essentially, as the algorithm learns more edges of the hypergraph of integer relations, future runs become more efficient. This also means that the algorithm can accept a hypergraph of integer relations at any point (even if partially filled), using the given integer relations to save time.

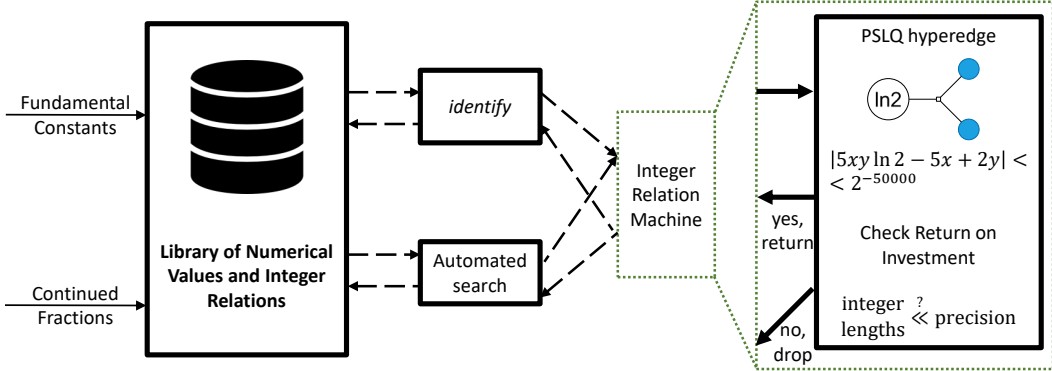

Figure 2: **Automated search of integer relations**. The process begins by collecting fundamental constants and continued fractions from the literature and organizing them into a database. Then, the algorithm checks subsets of constants for polynomial relations using PSLQ. We identify that a relation is significant by a high Return on Investment (RoI), as described in section 3. Such events are generally extremely rare, and each one is returned to the database and saved, enriching the hypergraph and adding to our knowledge about mathematical constants and their relations. Each such discovered relation also saves time on future runs of PSLQ. In addition, our novel *identify* utility (see section 4) allows for manually adding constants and continued fractions to the database regardless of the automated search.

# 3 IDENTIFYING INTEGER RELATIONS BETWEEN MATHEMATICAL CONSTANTS USING PSLQ

PSLQ is a numerically stable integer relation algorithm (Ferguson and Bailey, 1992). This algorithm uses a partial sum of squares scheme to manipulate a lattice, in a similar manner to the PSOS algorithm (Bailey and Ferguson, 1998), using LQ matrix decomposition (transposed QR decomposition). Follow-up works on PSLQ applied it in a wide range of settings (Bailey and Broadhurst, 2001). PSLQ accepts a vector of real numbers $x_1, x_2, ...$, and returns integers $a_1, a_2, ...$, such that $a_1 x_1 + a_2 x_2 + ... = 0$ within user-set tolerance, maximum number of steps, and maximum absolute value for all $a_i$ (or nothing if no such relation exists). Ordinarily, any of the three user-set parameters can terminate the algorithm's calculations, but in this section, we demonstrate how and why we run PSLQ only until tolerance is exhausted, and present a novel heuristic for deciding if its result is significant, which we have termed Return on Investment (RoI).

From a purely theoretical perspective, integer relations are either true or false, with the latter providing no insight towards the former. That is, if a certain integer relation is accurate to 100 digits of accuracy, but is then found to be incorrect at the 101st digit, it does not imply that any "similar" integer relation will be fully accurate. To identify promising relations despite the finite precision, we rely on a perspective motivated by data compression and statistics: Given $d$ binary digits, one can express about $2^{1+d}$ integers (including negatives). Thus, if our working precision is $d$ binary digits, and an integer relation algorithm yielded numbers with $d$ binary digits, the result is most likely a false positive. With a tighter analysis, if one splits this reasoning into $n$ integers, each with $d_1, d_2, ..., d_n$ binary digits, then the total amount of options for all integers is about $2^{1+d_1} \cdot 2^{1+d_2} \cdot ... = 2^{n+d_1+d_2+\cdots}$. As such, for precision of $d$ binary digits, any integer relation algorithm yielding integers with $d_1, d_2, ...$ binary digits such that $n + d_1 + d_2 + ... \approx d$ is most likely a false positive.

Based on the above analysis, we introduce and apply the concept of Return on Investment (RoI) to analyse the validity of integer relations: Given an integer relation with a precision of $d$, on $n$ integers each with $d_1, d_2, ...$ binary digits, the RoI is defined as $\frac{d}{n+d_1+d_2+...}$. With this definition in mind, it is reasonable to state that the higher the RoI, the more likely an integer relation is to be true. This assertion matches theory, as in that case, if an integer relation is theoretically true, then the RoI grows to $\infty$ as the working precision increases.

Experimenting with the PSLQ algorithm to determine an empirical lower bound for RoI, beyond which integer relations can be considered significant, shows that an RoI of 2 seems to comfortably separate random results from false positives, with a generous margin of error (see figure 3). As such, this has been decided as the lower cutoff for integer relations generated by PSLQ to be considered significant as part of the algorithm in section 2.1. In order to further show how general RoI can be, we performed a similar experiment using a different integer relation algorithm called LLL (Lenstra et al., 1982), and described its findings in appendix D.

## 3.1 CONTINUED FRACTION CALCULATION AND CONVERGENCE

Motivated by previous works of the Ramanujan Machine (Raayoni et al., 2021; Elimelech et al., 2024), generalized continued fractions (represented as $\mathcal{C}$-transforms) are a highly expressive framework of generating constants. One of the challenges of working with such mathematical formulas is that it is not always known in advance whether a certain formula converges to a limit. Nevertheless, our numerical experiments suggest that the asymptotics of each sequence determines its convergence properties and speed, leading to the following conjecture and analysis. This analysis aids in populating our database of continued fractions by rejecting formulas that do not convergence and helping with numerical identification tasks (see section 4). We propose the following conjecture, which generalizes on previous works (e.g., (Raayoni et al., 2021; Ben David et al., 2024)), providing the complete convergence conditions that we use when given an arbitrary $\mathcal{C}$-transform:

**Conjecture 1** *Let $f_n$ be a real sequence. The formula $\mathcal{C}[f_n]$ converges in the following cases, at a rate provided by the error $\varepsilon_n := |1 + f_1/(1 + f_2/(\cdots + f_n))) - \mathcal{C}[f_n]|$.*

- *If $f_n = O(n^k)$ for some $k < 0$, then $\varepsilon_n = O((n!)^k)$*

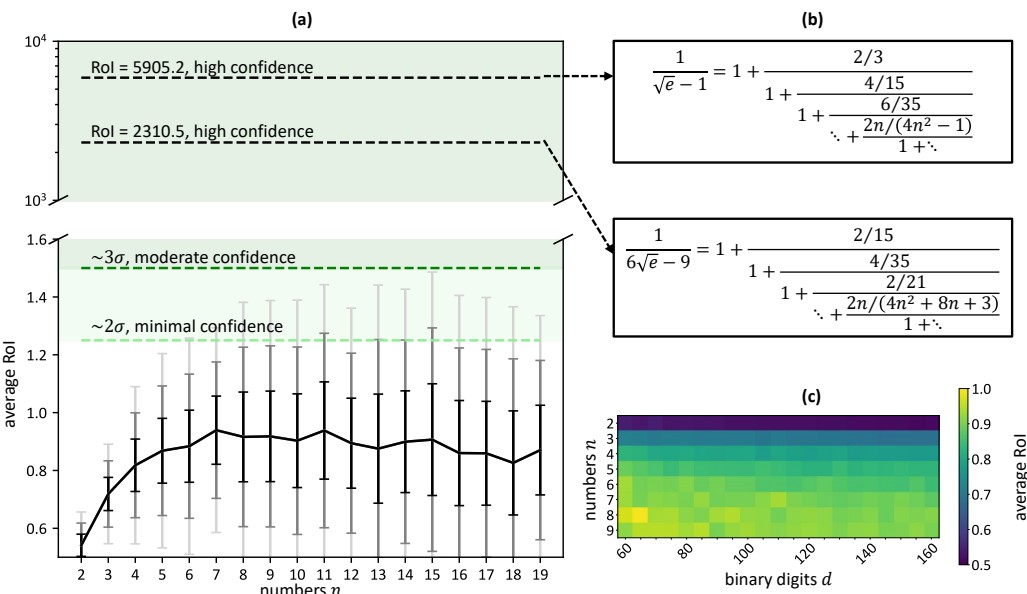

Figure 3: **Experimental analysis of the Return on Investment (RoI) property, showing its use for identifying integer relations.** (a) For each $n$, we ran PSLQ $100$ times with a pre-selected binary precision of $50 + 5n$. We present the average RoI for each $n$. The standard deviation for each $n$ is presented as fading errorbars, with the half length of each darkest error bar being equal to one standard deviation. The lighter dashed line is the constant RoI of $1.25$ and the darker dashed line is the constant RoI of $1.5$, which the plot suggests are viable options for minimum RoI for filtering integer relations. (b) Sampled formulas are listed with their RoI on panel *(a)*. Thanks to their high precision, their RoI is much greater than our recommended RoI cutoff. (c) For each $d, n$, we ran PSLQ $100$ times until tolerance (equal to $65\%$ of the working precision, see appendix B for more details), each with $n$ numbers between $0$ and $1$, each with $d$ uniformly random binary digits. Then, we present the average RoI across all $100$ runs for each $d, n$. For a fixed $n$, the average RoI is close to constant in $d$.

- *If $1 + 4f_n = C + o(1)$ for some constant $C \neq 1$, then $\varepsilon_n = O\left(\left|\frac{1+\sqrt{C}}{1-\sqrt{C}}\right|^{-n}\right)$*

- *If $1 + 4f_n = Cn^k + o(n^k)$ for some $k \in (0, 1]$, then $\varepsilon_n = O\left(e^{-4\sqrt{n/C}}\right)$*

- *If $1 + 4f_n = O(n^k)$ with $k < 0$ or $k \in (1, 2]$, then $\varepsilon_n = O(n^m)$ for some $m < 0$.*

*Otherwise, $\mathcal{C}[f_n]$ does not converge.*

Another challenge we face when analyzing continued fraction formulas is explicitly determining the limit of a converging $\mathcal{C}$-transform. In lieu of this, if a $\mathcal{C}$-transform converges then it is sufficient to stop at some finite evaluation depth and use the resulting convergents to calculate an approximation of the limit. Then, though it would be beneficial to know the exact error of the resulting approximation, it is much easier and readily available to calculate an error proxy based on the difference to the previous depth's approximation. Table 1 shows a few examples of this procedure, which also showcase the convergence model in action.

## 4 THE RAMANUJAN LIBRARY: PROVIDING PUBLIC ACCESS TO THE DATABASE OF INTEGER RELATIONS AND MATHEMATICAL CONSTANTS

The code we have written and the library we have curated are open-source and publicly-accessible,[1] including the novel database of mathematical constants and the hypergraph of integer relations.

---

[1]https://github.com/RamanujanMachine/LIReC

Table 1: **The predicted vs measured convergence properties of selected continued fractions in their canonical forms: error analysis.** Without prior knowledge, it is not possible to know exactly how inaccurate a given $\mathcal{C}$-transform's approximations are. Thankfully, the error predicted by the convergence model usually provides a good estimate of the error proxy, which is usually a lower bound on the true error. Both the error proxy and the true error are measured in terms of how many decimal digits agree between the approximation and the true limit. Each $\mathcal{C}$-transform here was either taken from the literature, in which case its limit is known, or otherwise found numerically, and then its limit is denoted with an asterisk and awaits proof. Note the lack of predicted error for $\mathcal{C}[n^2]$, due to no known formula for such $\mathcal{C}$-transforms (see conjecture 1).

| $\mathcal{C}$-transform | Evaluation depth | Predicted error | Error proxy | Limit | True error |
|---|---|---|---|---|---|
| $\mathcal{C}[1/n]$ | $2^{10}$ | 2640 | 2644 | $e - 1$ | 2647 |
| $\mathcal{C}[1]$ | $2^{10}$ | 428 | 427 | $(1 + \sqrt{5})/2$ | 428 |
| $\mathcal{C}[n]$ | $2^{10}$ | 27 | 26 | $\frac{1}{\sqrt{\frac{\pi e}{2}}\,\mathrm{erfc}\frac{1}{\sqrt{2}}}$ | 26 |
| $\mathcal{C}[n^2]$ | $2^{20}$ | N/A | 5 | $1/\ln 2$ (*) | 5 |

Table 2: **Sampled formulas from the hypergraph of integer relations**. Each formula here is taken from figure 4. As an example, consider how the first relation can be found using PSLQ: Assuming a polynomial with degree $2$ and order $1$, and given Catalan's constant $G$ and $\mathcal{C}\left[\frac{-2n^4}{9n^4-3n^2+1}\right]$, PSLQ can find integers $n_1, n_2, n_3, n_4$ such that $n_1\mathcal{C}\left[\frac{-2n^4}{9n^4-3n^2+1}\right]G + n_2\mathcal{C}\left[\frac{-2n^4}{9n^4-3n^2+1}\right] + n_3 G + n_4 = 0$. In this case, one can find $n_1 = 2, n_2 = 0, n_3 = 0, n_4 = -1$, and given that both $\mathcal{C}\left[\frac{-2n^4}{9n^4-3n^2+1}\right]$ and $G$ are known to high precision, this is a signal that an integer relation exists. The other relations here can be captured in a similar way, using more complex integer polynomials.

| Relation | Icon |
|---|---|
| $\mathcal{C}\left[\frac{-2n^4}{9n^4-3n^2+1}\right] = \frac{1}{2G}$ |  |
| $\ln 2 = \frac{1}{\mathcal{C}\left[\frac{n^2}{4-16n^2}\right]} - \frac{2}{5\mathcal{C}\left[\frac{n^2}{25-100n^2}\right]}$ |  |
| $\frac{\pi e}{2} = \left(\frac{1}{\mathcal{C}[n]} + \mathcal{C}\left[\frac{1}{2n}\right]\right)^2$ |  |
| $\mathcal{C}\left[\frac{-(n+4)(n+2)^2(2n+1)}{9n^4+84n^3+259n^2+294n+95}\right] = \frac{296-192\pi+180\zeta(2)}{1444-912\pi+855\zeta(2)}$ |  |

Our code also contains specialized algorithms for working with the hypergraph and for testing new candidate constants and candidate formulas. Access to the database from our code is established using *psycopg2* and *sqlalchemy* (Bayer, 2012). Our code contains for example the automated search algorithm described in section 2.1, along with additional utilities like the $\mathcal{C}$-transform calculator and the numerical identification suite we call *identify*, which are described in this section.

The computation of continued fractions is fairly straightforward thanks to the recursive relation mentioned in appendix A, and so the $\mathcal{C}$-transform calculator was developed to allow us to compute $\mathcal{C}$-transforms to arbitrary depths. The calculator utilizes three libraries for its computation: *sympy* (Meurer et al., 2017), *gmpy2*, and *mpmath* (mpmath development team, 2023). Using the above libraries, computation up to a depth of 1 million can be achieved in a few minutes at most on a typical personal computer, for most continued fractions. Using the error proxy and convergence model mentioned in section 3.1, we also give a preemptive analysis of any given $\mathcal{C}$-transform of a rational function, which is also useful to warn of $\mathcal{C}$-transforms that are expected to not converge

or to have a slow rate of convergence. This preemptive analysis also allows the user to specify their desired error proxy instead of having to guess an explicit depth in advance. Finally, when operating on $\mathcal{C}$-transforms of rational functions specifically, the calculator automatically shifts the given function so as to avoid its singularities and zeroes, resulting in an equivalent $\mathcal{C}$-transform up to some number of steps forwards or backwards (see figure 2 for a demonstration of this equivalence).

The integer relation machine depicted in figure 2 has two uses, one of which has been described in section 2.1, and the other is *identify*: a novel numerical identification tool.[2] The input to *identify* is an array of decimal expansions representing one or more constants and/or their implicit $\mathcal{C}$-transforms. The output is the result of putting all values into the integer relation machine to attempt to extract meaningful relations. If enabled, *identify* can also automatically fetch specific mathematical constants or $\mathcal{C}$-transforms from our database and match them against its input, informing the user of what the database is familiar with, if applicable. If at any point, *identify* is given a $\mathcal{C}$-transform that the database is unfamiliar with, *identify* will then automatically upload it to the database. Similarly, if a new relation is found in the process, it will also be uploaded to the database.

## 5 SELECTED DISCOVERED RELATIONS BETWEEN MATHEMATICAL CONSTANTS

The novel algorithm described in section 2.1 has yielded results which we summarize here. These results were found after running the algorithm on a smaller scale (an 8-core AWS machine) for several months. The total compute time of our results is, thus, about 16 compute months. However, since the algorithm is embarrassingly parallel, runtimes of the algorithm in practice go down with the number of CPU cores used. This shows the potential of our algorithm for running on much larger compute resources. Figure 4 shows 118 of the relations found, of which 43 were known in the literature and 75 are novel to the best of our knowledge. Figure 2 shows some example relations. Appendix F catalogues all relations in detail. These results were found after starting with a completely disconnected hypergraph, containing the constants of interest, and no initial relations. Here we present notable discovered relations involving $\pi$, $e$, $ln2$ and the Lemniscate constants.

We recall the following formula, attributed to Ramanujan (Berndt et al., 1999) (using the notation defined in equation 1, section 2):

$$\sqrt{\frac{\pi e}{2}} = \frac{1}{\mathcal{C}[n]} + \frac{1}{2\mathcal{C}[(1-2n)/(4n(n+1))] - 1}$$

The search job unearthed a family of conjectures that generalizes the second $\mathcal{C}$-transform in this relation, given in table 3.

Next, we examined the family of constants $\mathcal{C}[n^2/(k^2(1-4n^2))]$ for integers $k \geq 1$, first noted in Ben David et al. (2024) (table 4 column 1). When they were first discovered, these $\mathcal{C}$-transforms were known to converge but their limits were unknown. Our search algorithm connected three of these $\mathcal{C}$-transforms to $ln2$ in pairs:

$$ln2 = \frac{1}{2\mathcal{C}[n^2/(4(1-4n^2))]} + \frac{1}{7\mathcal{C}[n^2/(49(1-4n^2))]} =$$

$$= \frac{2}{5\mathcal{C}[n^2/(25(1-4n^2))]} + \frac{2}{7\mathcal{C}[n^2/(49(1-4n^2))]} =$$

$$= \frac{1}{\mathcal{C}[n^2/(4(1-4n^2))]} - \frac{2}{5\mathcal{C}[n^2/(25(1-4n^2))]}$$

Thanks to the high precision to which these equations have been verified, our Return on Investment (RoI) measure (section 3) quantifies how unlikely they are to be false conjectures. These conjectures are also listed in appendix F. Later investigation has revealed:

**Conjecture 2** *For all $k \geq 1$,* $\mathcal{C}\left[\frac{n^2}{k^2(1-4n^2)}\right] = \frac{2/k}{\ln(k+1)-\ln(k-1)}$.

---

[2]See README in the Github link, or the tutorial hosted at:
https://colab.research.google.com/drive/1PXAn4FwTHn0YQIBNDmOSIHWensqKetcU

Table 3: **Discovered Ramanujan-like formulas for** $\sqrt{\pi e}$. For brevity, we denote $R_2 = \sqrt{\frac{\pi e}{2}}\mathrm{erf}\frac{1}{\sqrt{2}}$. The rows in this table represent the 8 formulas we discovered for $\sqrt{\pi e}$, generalizing Ramanujan's original formula that was so far unique. We proved the first 4 rows using transformations on Ramanujan's original formula. The latter 4 rows are still unproven (yet equivalent, such that proving any one of them will prove all four). The last row of each set of four is an infinite family of formulas: for any integer $k \geq 0$ there exist integers $\alpha, \beta, \gamma, \delta$ that satisfy an equality between the two sides.

| $\mathcal{C}$-transform | Limit |
|---|---|
| $\mathcal{C}[(1-2n)/(4n(n+1))]$ | $(R_2+1)/2R_2$ |
| $\mathcal{C}[(1-2n)/(4(n+1)(n+2))]$ | $(2R_2+1)/4$ |
| $\mathcal{C}[(1-2n)/(4(n+2)(n+3))]$ | $(R_2+1)/(6R_2-1)$ |
| $\mathcal{C}[(1-2n)/(4(n+k)(n+k+1))]$ | $(\alpha R_2 + \beta)/(\gamma R_2 + \delta)$ |
| $\mathcal{C}[n/(2n(n+1))]$ | $1/(2R_2-1)$ |
| $\mathcal{C}[(n+1)/(2n(n+1))]$ | $R_2$ |
| $\mathcal{C}[(n+2)/(2n(n+1))]$ | $(2R_2+1)/(R_2+1)$ |
| $\mathcal{C}[(n+k)/(2n(n+1))]$ | $(\alpha R_2 + \beta)/(\gamma R_2 + \delta)$ |

As a final example of notable results, we compared the performance of *identify* with commercial numerical identification tools such as Wolfram Alpha [3]. Our *identify* method was successful in cases for which Wolfram Alpha was unsuccessful in identifying the constant. This was the case for formulas such as $\mathcal{C}[-(2n+3)^2/(18n(n+1))]$ (as an example of a larger family of $\mathcal{C}$-transforms). Executing *identify* found that the limit is $\frac{5A+6B}{5A-9B}$ (to 50 digits of accuracy initially, and further thousands upon reconfirming), where $A, B$ are respectively the first and second Lemniscate constants.

## 6 LIMITATIONS AND OUTLOOK

Our work presented a successful approach to automated formula discovery in number theory. The numerical nature of our algorithms means that results are not theorems, but rather conjectures awaiting proofs. In many cases, the discovered relations between formulas led to patterns that enabled generalizing formulas into families. Some of these families are connected such that proving one formula proves the entire family. We provided proofs for selected cases (shown in appendix E), hinting that the other discovered formulas can also be proven. In a similar vein, our findings reveal the complete rules of convergence of general continued fractions, yet these discovered rules lack a formal proof for many of the special cases. Future work by number theorists can follow on our groundwork to complete the proof and build the complete theory of convergence of continued fractions.

The Return on Investment (RoI) mechanism that we presented to quantify the success of the PSLQ algorithm can be directly extended to other integer relation algorithms such as LLL (Lenstra et al., 1982) (see also appendix D). This quantitative approach can contribute to the application of such algorithms in a wider range of fields, and will benefit from a thorough theoretical investigation.

Our approach and implementation can benefit from many improvements. The search algorithm was implemented in a way that allows for distributed computing. We are now adapting the algorithm to utilize additional computing power using the online distributing network BOINC (Anderson, 2004). Such large-scale computing efforts will be very promising in generating new, more elaborate formulas and interrelations. The increase in the number of interrelations will require corresponding methods to eliminate false positives, which could be accomplished automatically by retesting them over time with higher precision constants. Sufficient precision will eventually reveal each potential false positive. Moreover, the current automated search algorithm relies on a pre-selected, static set of constants, which directly determines which relations can be found. Methods of automatically adding "interesting" constants to the database can improve the potential for finding novel relations.

Looking forward, the publicly-available Ramanujan library that we developed can now serve number theorists for investigating mathematical constants with more powerful tools that are easily accessible. The hypergraph of integer relations may be viewed like a knowledge graph, and as a public resource, it can help open such areas of investigations to a broader community with different levels of expertise.

---

[3] https://www.wolframalpha.com/input?i=-0.34391017046397691140258013367681311419903292310618

By presenting fascinating connections between constants, we hope that this resource will invoke curiosity in younger audiences and encouraging mathematical experimentation as a path for learning mathematics. In its next generations, the Ramanujan library can be improved by augmenting the hypergraph with written explanations on constants and links to relevant papers, forming a public hub of knowledge on constants and their relations.

## ACKNOWLEDGEMENTS

This research is supported by the generosity of Eric and Wendy Schmidt by recommendation of the Schmidt Futures Polymaths program.

This research received software engineering support from the Georgia Institute of Technology's Scientific Software Engineering supported by Schmidt Sciences, as part of the Virtual Institute for Scientific Software (VISS) Program

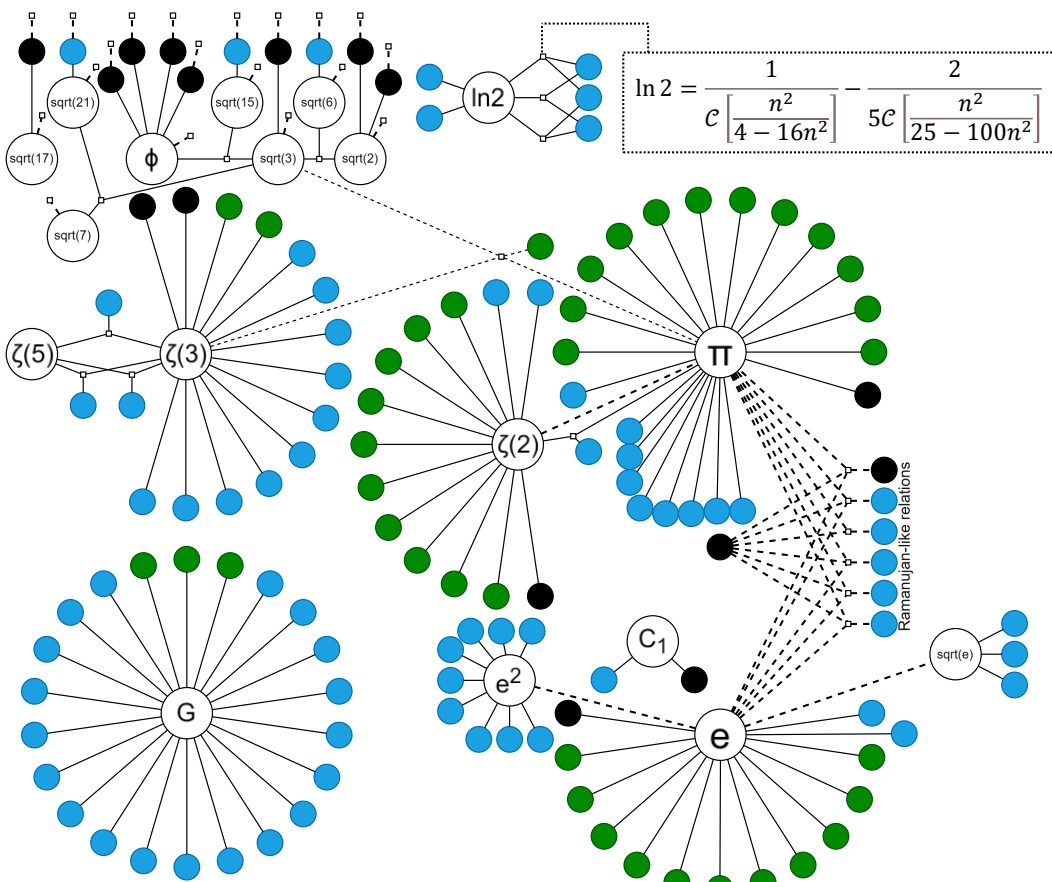

Figure 4: **The hypergraph of integer relations: each vertex is a constant, and each edge is a formula**. The hypergraph summarizes our automated discovery of relations between mathematical constants, presenting selected formulas from our database. This hypergraph does not show all discovered relations for clarity's sake. Empty circles denote constants (written inside). Colored circles denote continued fractions. Black denotes continued fractions whose limit is known in the broader literature. Green denotes continued fractions whose limit is found in (Raayoni et al., 2021). Blue denotes continued fractions whose limit was unknown before our work, to the best of our knowledge. We denote order-1 connections (Mobius-like) with solid lines, and higher-order connections (constants may appear squared, cubed, etc.) with dashed lines. Edges that connect more than two vertices are marked with small empty squares, denoting formulas that involve more than two constants of formulas (example in the inset at the top right). Each algebraic constant has an edge of size 1, also marked with a small empty square (placed at the top left). Such degenerate edges correspond to the minimal polynomial of the constant.

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

## A  ON GENERALIZED CONTINUED FRACTIONS

A generalized continued fraction, given two sequences $a_n, b_n$, is the following formal expression:

$$GCF[a_n, b_n] = a_0 + \cfrac{b_1}{a_1 + \cfrac{b_2}{a_2 + \cfrac{b_3}{\ddots}}}$$

In practice, one defines two sequences $p_n, q_n$ called the convergents, such that

$$a_0 + \cfrac{b_1}{a_1 + \cfrac{b_2}{a_2 + \cfrac{b_3}{\ddots + \cfrac{b_n}{a_n}}}} = \frac{p_n}{q_n}$$

and then both sequences satisfy a recurrence relation, summarized by the following matrix product:

$$\begin{bmatrix} 1 & a_0 \\ 0 & 1 \end{bmatrix} \prod_{i=1}^{n} \begin{bmatrix} 0 & b_i \\ 1 & a_i \end{bmatrix} = \begin{bmatrix} p_{n-1} & p_n \\ q_{n-1} & q_n \end{bmatrix}$$

The special case of $b_n \equiv 1$ is called a (regular) continued fraction. $\mathcal{C}$-transforms are the special case of $a_n \equiv 1$. Using basic transformations, any generalized continued fraction can be converted into this latter form.

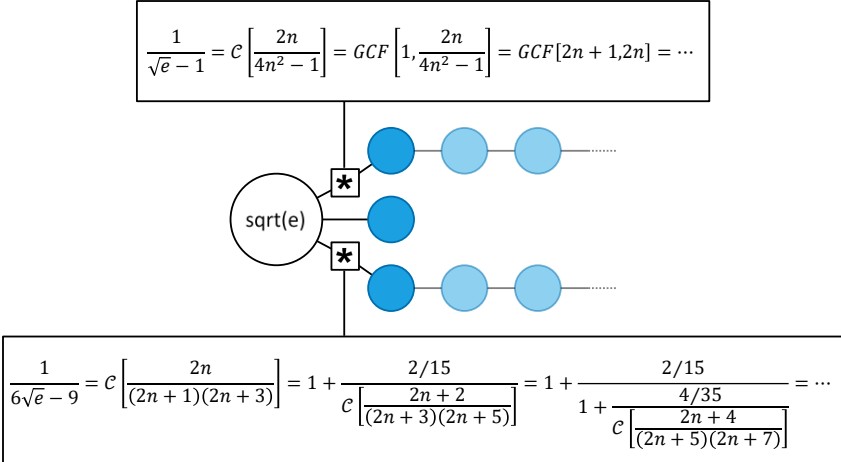

Figure 5: **Equivalences between continued fractions**. Expanding on figure 1, our representation of the continued fractions enables each integer relation to capture an infinite family of formulas. This redundancy is mostly eliminated by using the $\mathcal{C}$-transform. Additional equivalences still exist, as discussed in section 4 and shown by the second integer relation.

Our choice of $\mathcal{C}$-transforms arises from the following *equivalence transformation* property of continued fractions: Given any $GCF[a_n, b_n]$ and nonzero sequence $c_n$, the following equality holds:

$$a_0 + \cfrac{b_1}{a_1 + \cfrac{b_2}{a_2 + \cfrac{b_3}{\ddots}}} = a_0 + \cfrac{c_1 b_1}{c_1 a_1 + \cfrac{c_1 c_2 b_2}{c_2 a_2 + \cfrac{c_2 c_3 b_3}{\ddots}}}$$

Or, more compactly, $c_0 GCF[a_n, b_n] = GCF[c_n a_n, c_{n-1} c_n b_n]$ (see figure 5 for more concrete examples). This equality holds even if neither side converges, in which case the convergents still coincide for all $n$. As such, if $a_n \neq 0$, the choice of $c_n = \frac{1}{a_n}$ yields $\frac{1}{a_0} GCF[a_n, b_n] = \mathcal{C}[b_n/(a_{n-1} a_n)]$.

## B   REBOUNDING DETECTION

Since we have elected to nullify two of the three stopping conditions of PSLQ, we must consider whether or not the remaining stopping condition can always be achieved. Namely, can every run of PSLQ terminate when only stopping once tolerance has been achieved. Even with a fairly standard choice of $75\%$ tolerance, the answer is no, at least for the implementation we use. However, our experiments show that all such cases appear to exhibit the same general behavior: PSLQ first gradually increases the precision of the integer relation as usual, and at some point before reaching tolerance, it reverses course and starts reducing the precision (see figure 6 for an example run).

Our implementation of PSLQ includes a failsafe that detects this behavior: As PSLQ iterates in the main algorithm, remember $p_{best}$ the best precision ever obtained in the current run. If at any point $p_{current}$ the current precision satisfies $2p_{current} < p_{best}$, abort the algorithm and return no integer relation. However, this mechanism is only enabled after an initial grace period of 100 steps. This grace period was chosen since it was observed that during the phase where the precision increases, it does not always increase monotonically, and can fluctuate.

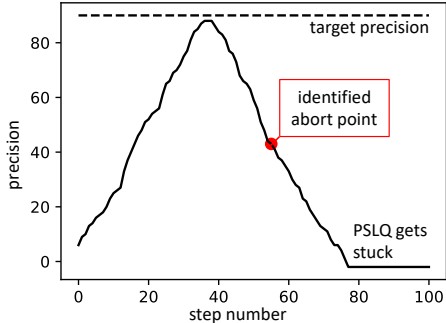

Figure 6: **Example run of PSLQ showcasing the rebounding phenomenon**. The y axis represents the precision of the best integer relation that PSLQ finds at each step number, which must reach the dashed line to terminate. The red dot is where the failsafe would have triggered, except that it is being prevented by the grace period to enable computing this example.

## C   TIME COMPLEXITY OF THE PSLQ ALGORITHM

Given $x$ a vector of $m$ real numbers, and assuming the minimal euclidean norm of all integer relations on it is $M_x \in \mathbb{R}$, running PSLQ to retrieve an integer relation on $x$ has a time complexity of $O(m^4 + m^3 \log M_x)$ (see (Ferguson et al., 1999) corollary 2). To estimate the time complexity of our runs of PSLQ, we substitute $M_x$ with a constant based on the working precision. When applying PSLQ on a large scale, choosing an upper bound for the precisions of the constants involved prevents this $M_x$ term from dominating the runtime, and so we can asymptotically treat this as a constant, leaving us with a time complexity of $O(m^4)$.

Since we run PSLQ to detect specifically polynomial relations, we should evaluate $m$ in terms of the amount of constants $n$, the maximal degree $d$, and the order $o$ of the relation, as these dictate how many monomials we expect the polynomial to have, and in turn determine $m$. If we ignore the limitation on the order, counting all possible monomials of a polynomial on $n$ variables with degree $d$ is identical to counting all multisets of size $d$ on $n + 1$ items, each of which represents either one of the variables, or a dummy variable that completes the monomial to degree $d$. This has a closed form solution based on the stars and bars method:

$$m = \binom{n + 1 + d - 1}{d} = \binom{n + d}{d} = \frac{(n+d)!}{d!n!}.$$

When including the limitation on the order, one obtains a smaller value than this through a more delicate analysis, but to the best of our knowledge, no closed form solution exists. Regardless, this means that each run of PSLQ will have a time complexity of $O\left(\binom{n+d}{d}^4\right)$.

# D THE GENERALIZEABILITY OF RETURN ON INVESTMENT (RoI)

The RoI measure that we introduced in section 3 is defined on numerically-conjectured integer relations. Our work relies on the PSLQ algorithm, which is one of many integer relation algorithms, and so it is natural to ask if RoI can be applied to other integer relation algorithms and reach the same conclusions. Here we motivate a positive answer to this question by demonstrating the same experiment shown in section 3 (specifically figure 3), but performed on the LLL algorithm instead (Lenstra et al., 1982).

Figure 7, which is structured in an identical manner to figure 3, demonstrates a nearly identical experiment, except the PSLQ algorithm is replaced by the LLL algorithm. This experiment used the *olll* python package for an implementation of LLL. The result shows how the algorithms are similar yet different in terms of RoI: as $n$ increases, both algorithms exhibit greater average RoI, up to some apparent upper limit less than $1.5$. However, PSLQ showed greater standard deviation than LLL, which is also visible as greater noise on each figure's *(c)* panel.

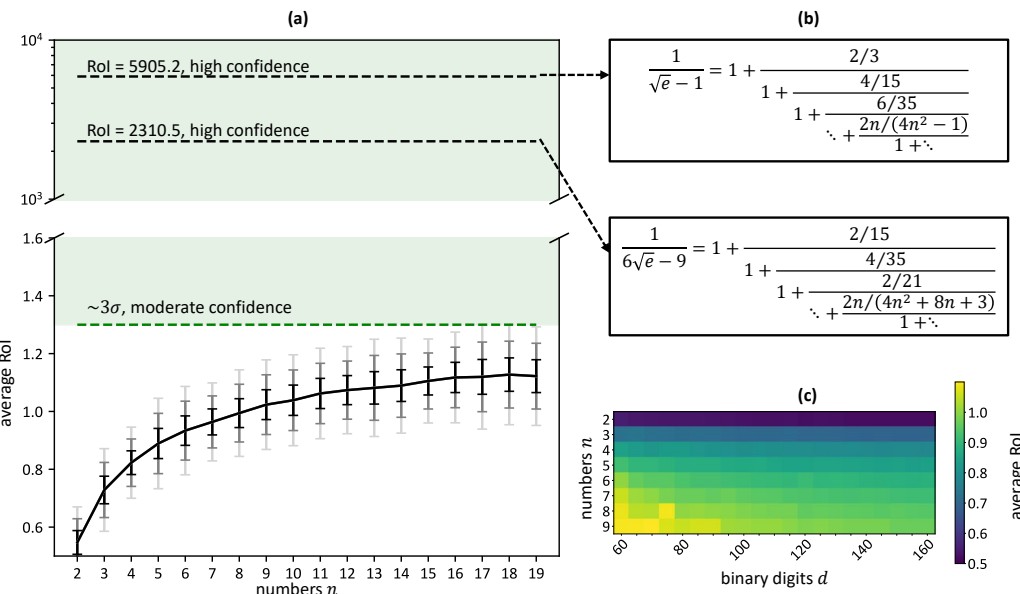

Figure 7: **Experimental analysis of the Return on Investment (RoI) property, demonstrated on the LLL algorithm.** (a) For each $n$ we ran LLL $100$ times, with a preselected binary precision of $50 + 5n$, and present the average RoI for each $n$. The standard deviation for each $n$ is presented as fading errorbars, with the half length of each darkest error bar being equal to one standard deviation. The dashed line is the constant RoI of $1.3$, which the plot suggests is a viable option for minimum RoI for filtering integer relations from LLL. (b) Sampled formulas listed with their RoI on panel *(a)*. Thanks to their high precision, their RoI is much larger than our recommended RoI cutoff. (c) For each $d, n$, we ran LLL $100$ times, each with $n$ numbers between $0$ and $1$, each with $d$ uniformly random binary digits. Then, we present the average RoI across all $100$ runs for each $d, n$. For a fixed $n$, the average RoI is close to constant in $d$.

# E PROVEN FORMULAS

After having found the formulas shown in the hypergraph (figure 4), we proved some of them, and we list these proven formulas here. We also plan joint works with mathematicians that focus on the proofs of the formulas and their potential impact in the relevant fields of mathematics, to be submitted to mathematics journals. This effort complements our main contribution in this work, which is concerned with the automated generation of the formulas.

In total, we proved 47 formulas of the form $a_0 + b_1/(a_1 + b_2/(... + b_n/(a_n + ...)))$, of which 22 we showcase in this table. 6 of the formulas can be generalized to three parametric families,

presented in lines 1-9. Lines 10-25 describe additional proven formulas. The proofs rely on (1) inverting Euler's continued fraction formula with parameters $h_1, h_2, f$ to (2) create an infinite sum, which we (3) translate into a hypergeometric-function form, and (4) complete the proof using known hypergeometric identities.

Table 4: **Selected list of formulas that we proved.**

| $a_n$ | $b_n$ | $h_1$ | $h_2$ | $f$ | Known limit of resulting infinite sum |
|---|---|---|---|---|---|
| $\omega+1$ | $n^2+\omega n$ | $-n$ | $n+\omega$ | $1$ | $(1+\omega)({}_2F_1(1,1,\omega+2,-1)^{-1}-1)$ |
| $5$ | $n^2+4n$ | $-n$ | $n+4$ | $1$ | $\frac{-12}{131+192\log 2}$ |
| $4$ | $n^2+3n$ | $-n$ | $n+3$ | $1$ | $\frac{3}{-16+24\log 2}$ |
| $\omega$ | $(\omega n+1)^2$ | $-\omega n-1$ | $\omega n+1$ | $1$ | $(\omega+1)({}_2F_1(1,\frac{\omega+1}{\omega},\frac{2\omega+1}{\omega},-1)^{-1}-1)$ |
| $2$ | $4n^2+4n+1$ | $-2n-1$ | $2n+1$ | $1$ | $\frac{\pi}{4-\pi}$ |
| $1$ | $n^2+2n+1$ | $-n-1$ | $n+1$ | $1$ | $\frac{\log(2)}{1-\log 2}$ |
| $\omega$ | $n^2+(\omega+1)n+\omega$ | $-n-1$ | $n+\omega$ | $1$ | $(\omega+1)({}_2F_1(1,2,\omega+2,-1)^{-1}-1)$ |
| $3$ | $n^2+4n+3$ | $-n-1$ | $n+3$ | $1$ | $-1+\frac{4}{34-48\log 2}$ |
| $2$ | $n^2+3n+2$ | $-n-1$ | $n+2$ | $1$ | $-1+\frac{3}{9-12\log 2}$ |
| $5$ | $n^2+2n$ | $n$ | $n+2$ | $n+\frac{3}{2}$ | $\frac{-2}{17-24\log 2}$ |
| $5$ | $n^2+4n+3$ | $-n-1$ | $n+3$ | $n+\frac{5}{2}$ | $\frac{-3}{5}+\frac{28}{5(\frac{-1162}{3}+560\log 2)}$ |
| $4$ | $n^2+n$ | $-n$ | $n+1$ | $n+1$ | $\frac{4}{12-16\log 2}$ |
| $4$ | $n^2+3n+2$ | $-n-1$ | $n+2$ | $n+2$ | $\frac{-1}{2}+\frac{9}{2(-99+144\log 2)}$ |
| $4$ | $4n^2+4n$ | $-2n$ | $2n+2$ | $1$ | $\frac{4}{-2+4\log 2}$ |
| $3$ | $n^2$ | $-n$ | $n$ | $n+\frac{1}{2}$ | $\frac{3}{3-3\log 2}$ |
| $3$ | $n^2+2n+1$ | $-n-1$ | $n+1$ | $n+\frac{3}{2}$ | $\frac{-1}{3}+\frac{10}{3(-20+30\log 2)}$ |
| $3$ | $n^2+4n+4$ | $-n-2$ | $-n+2$ | $n+\frac{5}{2}$ | $\frac{-6}{5}-\frac{21}{5(\frac{147}{2}+105\log 2)}$ |
| $1$ | $n^2+4n+4$ | $n+2$ | $-n-2$ | $1$ | $-2+\frac{3}{\frac{-3}{2}+3\log 2}$ |
| $4$ | $4n^2-1$ | $-2n+1$ | $2n+1$ | $1$ | $\frac{3}{\frac{-3}{2}+\frac{3\pi}{4}}+1$ |
| $2$ | $4n^2-4n-1$ | $-2n+1$ | $2n-1$ | $1$ | $\frac{4}{\pi}+1$ |
| $5$ | $4n^2+2n-2$ | $-2n-1$ | $2n+2$ | $1$ | $\frac{4}{\frac{-20}{3}+\frac{16\sqrt{2}}{3}}+1$ |
| $5$ | $4n^2+2n$ | $-2n-1$ | $2n$ | $n+\frac{3}{4}$ | $3+2\sqrt{2}$ |
| $3$ | $4n^2-2n$ | $-2n+1$ | $2n$ | $1$ | $2+\sqrt{2}$ |
| $2n^2+2n+1$ | $-n^4$ | $n^2$ | $n^2$ | $1$ | $\frac{1}{\zeta(2)}$ |
| $2n+1$ | $n^4$ | $-n^2$ | $n^2$ | $1$ | $\frac{2}{\zeta(2)}$ |

## F  DETAILED INTEGER RELATIONS FROM THE HYPERGRAPH

This section provides the full data underlying the hypergraph (figure 4). Recall that according to appendix A, each of the integer relations presented here can be connected to infinitely many generalized continued fractions, and the $\mathcal{C}$-transform is chosen as a representative for them all. Thus, every vertex in the hypergraph could represent infinite formulas that we consider trivially equivalent. This reasoning motivated the use of the $\mathcal{C}$-transform as the canonical representative that defines the vertex. The edges in the hypergraph provide additional (non-trivial) equivalences. Some of the formulas and relations shown here can be found in the literature, and others are novel, to the best of our knowledge (as denoted in the left-most column). Every integer relation has been rearranged for clarity, though binary precision and RoI (rounded down) are still computed in terms of the original integer relation.

| Novelty | Integer relation | Precision | RoI |
|---|---|---|---|

| | | | |
|---|---|---|---|
| Known | $\mathcal{C}\left[\frac{-n^6}{1156n^6-765n^4+219n^2-25}\right] = \frac{6}{5\zeta(3)}$ | 53147 | 6643.3 |
| Known | $\mathcal{C}\left[\frac{-n^6}{36n^6-21n^4+7n^2-1}\right] = \frac{8}{7\zeta(3)}$ | 53146 | 5905.1 |
| Known | $\mathcal{C}\left[\frac{-16n^6}{100n^6-45n^4+21n^2-4}\right] = \frac{6}{7\zeta(3)}$ | 53147 | 6643.3 |
| New | $\mathcal{C}\left[\frac{-n^{10}}{4n^{10}+63n^8+264n^6-154n^4+57n^2-9}\right] = \frac{2}{6\zeta(5)-6\zeta(3)+3}$ | 160 | 11.4 |
| New | $\mathcal{C}\left[\frac{-n^{10}}{4n^{10}+63n^8+296n^6+158n^4+137n^2-49}\right] = \frac{2}{14\zeta(5)+42\zeta(3)-63}$ | 162 | 7.3 |
| New | $\mathcal{C}\left[\frac{-n^{10}}{4n^{10}+143n^8+1364n^6+286n^4+517n^2-169}\right] = \frac{64}{832\zeta(5)+2288\zeta(3)-3549}$ | 234 | 5.2 |
| New | $\mathcal{C}\left[\frac{-n^6}{4n^6+99n^4+651n^2-169}\right] = \frac{8}{104\zeta(3)-117}$ | 197 | 9.3 |
| New | $\mathcal{C}\left[\frac{-n^6}{4n^6+195n^4+2451n^2-625}\right] = \frac{216}{5400\zeta(3)-6275}$ | 269 | 7.2 |
| New | $\mathcal{C}\left[\frac{4n^6-2n^5}{9n^6+33n^5+25n^4-13n^3-12n^2+2n+2}\right] = \frac{5}{4\zeta(3)}$ | 53147 | 6643.3 |
| New | $\mathcal{C}\left[\frac{-n^6}{4n^6+35n^4+91n^2-25}\right] = \frac{1}{5\zeta(3)-5}$ | 122 | 12.2 |
| New | $\mathcal{C}\left[\frac{-4n^6}{16n^6+572n^4+5332n^2-1369}\right] = \frac{6750}{874125\zeta(3)-1043992}$ | 226 | 4 |
| New | $\mathcal{C}\left[\frac{-4n^6}{16n^6+252n^4+1092n^2-289}\right] = \frac{54}{3213\zeta(3)-3808}$ | 156 | 4.7 |
| New | $\mathcal{C}\left[\frac{-n^6}{4n^6+323n^4+6643n^2-1681}\right] = \frac{1728}{70848\zeta(3)-83435}$ | 341 | 7.1 |
| New | $\mathcal{C}\left[\frac{-n^6}{4n^6+675n^4+28731n^2-7225}\right] = \frac{4800}{408000\zeta(3)-485639}$ | 487 | 9 |
| New | $\mathcal{C}\left[\frac{-n^6}{4n^6+483n^4+14763n^2-3721}\right] = \frac{216000}{13176000\zeta(3)-15622283}$ | 409 | 5.9 |
| Known | $\mathcal{C}\left[\frac{-2n^4-9n^3-12n^2-4n}{9n^4+48n^3+85n^2+56n+9}\right] = \frac{8}{54-27\zeta(2)}$ | 53146 | 2952.5 |
| Known | $\mathcal{C}\left[\frac{n^4}{121n^4-55n^2+9}\right] = \frac{5}{3\zeta(2)}$ | 53149 | 7592.7 |
| Known | $\mathcal{C}\left[\frac{-2n^4-13n^3-22n^2-8n}{9n^4+72n^3+189n^2+180n+45}\right] = \frac{16}{240-135\zeta(2)}$ | 53144 | 2214.3 |
| Known | $\mathcal{C}\left[\frac{-4n^4-6n^3}{25n^4+90n^3+111n^2+54n+10}\right] = \frac{9}{30\zeta(2)-40}$ | 53146 | 2952.5 |
| Known | $\mathcal{C}\left[\frac{-2n^4-5n^3-3n^2+n+1}{9n^4+36n^3+51n^2+30n+7}\right] = \frac{3\zeta(2)+2}{42-21\zeta(2)}$ | 53146 | 2797.1 |
| Known | $\mathcal{C}\left[\frac{-2n^4-9n^3-9n^2+n+3}{9n^4+60n^3+139n^2+130n+39}\right] = \frac{9\zeta(2)}{208-117\zeta(2)}$ | 53144 | 2415.6 |
| Known | $\mathcal{C}\left[\frac{-2n^4-7n^3-4n^2+4n}{9n^4+48n^3+85n^2+56n+9}\right] = \frac{8}{27\zeta(2)-36}$ | 53146 | 2952.5 |
| Known | $\mathcal{C}\left[\frac{-2n^4-11n^3-18n^2-4n+8}{9n^4+72n^3+213n^2+276n+133}\right] = \frac{12\zeta(2)-32}{456-285\zeta(2)}$ | 53142 | 1660.6 |
| Known | $\mathcal{C}\left[\frac{8n^4}{49n^4-21n^2+4}\right] = \frac{2}{\zeta(2)}$ | 53151 | 10630.2 |
| Known | $\mathcal{C}\left[\frac{-2n^4+n^3}{9n^4-3n^2+1}\right] = \frac{4}{3\zeta(2)}$ | 53149 | 7592.7 |
| Known | $\mathcal{C}\left[\frac{-4n^4+2n^3}{25n^4+10n^3-9n^2-2n+2}\right] = \frac{3}{2\zeta(2)}$ | 53150 | 8858.3 |
| New | $\mathcal{C}\left[\frac{-2n^4-9n^3-9n^2+n+3}{9n^4+48n^3+61n^2-8n-15}\right] = \frac{45\zeta(2)-48\pi+66}{225\zeta(2)-240\pi-370}$ | 53143 | 1062.8 |
| New | $\mathcal{C}\left[\frac{-n^4}{4n^4+24n^2+49}\right] = \frac{2}{14\zeta(2)-21}$ | 101 | 7.2 |
| New | $\mathcal{C}\left[\frac{-n^4}{4n^4+48n^2+169}\right] = \frac{18}{403-234\zeta(2)}$ | 138 | 5.5 |
| New | $5\ln2 = \frac{5}{\mathcal{C}\left[\frac{-n^2}{16n^2-4}\right]} - \frac{2}{\mathcal{C}\left[\frac{-n^2}{100n^2-25}\right]}$ | 53150 | 4831.8 |
| New | $14\ln2 = \frac{2}{\mathcal{C}\left[\frac{-n^2}{196n^2-49}\right]} + \frac{7}{\mathcal{C}\left[\frac{-n^2}{16n^2-4}\right]}$ | 53148 | 4429 |
| New | $35\ln2 = \frac{10}{\mathcal{C}\left[\frac{-n^2}{196n^2-49}\right]} + \frac{14}{\mathcal{C}\left[\frac{-n^2}{100n^2-25}\right]}$ | 53147 | 3126.2 |
| New | $\mathcal{C}\left[\frac{-n^2}{36n^2-9}\right] = \frac{2}{3\ln2}$ | 53150 | 8858.3 |

| New | $\mathcal{C}\left[\frac{-4n^4-6n^3}{16n^4+24n^3+117n^2+81n+238}\right]=\frac{3}{544\ln 2-374}$ | 108 | 4.5 |
|---|---|---|---|
| Known | $\mathcal{C}\left[\frac{-2n^4-10n^3-14n^2-6n}{9n^4+60n^3+139n^2+130n+39}\right]=\frac{6}{221-234G}$ | 53144 | 2415.6 |
| Known | $\mathcal{C}\left[\frac{-2n^4-4n^3}{9n^4+24n^3+13n^2-4n-3}\right]=\frac{2}{6G-3}$ | 53149 | 5314.9 |
| Known | $\mathcal{C}\left[\frac{-2n^4-8n^3-8n^2}{9n^4+48n^3+85n^2+56n+9}\right]=\frac{4}{54G-45}$ | 53146 | 2952.5 |
| New | $\mathcal{C}\left[\frac{-2n^4-6n^3}{9n^4-3n^2+1}\right]=\frac{720}{450G-299}$ | 53140 | 1714.1 |
| New | $\mathcal{C}\left[\frac{-2n^4-12n^3-24n^2-16n}{9n^4+48n^3+85n^2+56n+9}\right]=\frac{32}{18G+45}$ | 53148 | 2657.4 |
| New | $\mathcal{C}\left[\frac{-2n^4-14n^3-32n^2-24n}{9n^4+48n^3+85n^2+56n+9}\right]=\frac{64}{54G+39}$ | 53146 | 2415.7 |
| New | $\mathcal{C}\left[\frac{-2n^4-2n^3}{9n^4-3n^2+1}\right]=\frac{2}{2G-1}$ | 53149 | 6643.6 |
| New | $\mathcal{C}\left[\frac{-2n^4-12n^3-16n^2}{9n^4+72n^3+189n^2+180n+45}\right]=\frac{16}{450G-395}$ | 53143 | 2043.9 |
| New | $\mathcal{C}\left[\frac{-2n^4-14n^3-28n^2-16n}{9n^4+72n^3+189n^2+180n+45}\right]=\frac{32}{285-270G}$ | 53144 | 1968.2 |
| New | $\mathcal{C}\left[\frac{-2n^4-16n^3-40n^2-32n}{9n^4+72n^3+189n^2+180n+45}\right]=\frac{128}{255-90G}$ | 53145 | 2044 |
| New | $\mathcal{C}\left[\frac{-2n^4-4n^3}{9n^4-3n^2+1}\right]=\frac{24}{18G-11}$ | 53145 | 3126.1 |
| New | $\mathcal{C}\left[\frac{-2n^4-12n^3-24n^2-16n}{9n^4+72n^3+213n^2+276n+133}\right]=\frac{8}{1026G-931}$ | 53141 | 1897.8 |
| New | $\mathcal{C}\left[\frac{-2n^4-14n^3-30n^2-18n}{9n^4+84n^3+283n^2+406n+207}\right]=\frac{12}{1909-2070G}$ | 53141 | 1771.3 |
| New | $\mathcal{C}\left[\frac{-2n^4-8n^3-8n^2}{9n^4+24n^3+13n^2-4n-3}\right]=\frac{16}{18G-3}$ | 53147 | 3543.1 |
| New | $\mathcal{C}\left[\frac{-2n^4}{9n^4-3n^2+1}\right]=\frac{1}{2G}$ | 53151 | 10630.2 |
| New | $\mathcal{C}\left[\frac{-2n^4-10n^3-12n^2}{9n^4+24n^3+13n^2-4n-3}\right]=\frac{96}{90G-31}$ | 53144 | 2415.6 |
| New | $\mathcal{C}\left[\frac{-2n^4-6n^3-6n^2-2n}{9n^4+36n^3+51n^2+30n+7}\right]=\frac{1}{14-14G}$ | 53148 | 4429 |
| New | $\mathcal{C}\left[\frac{-2n^4-6n^3-4n^2}{9n^4+24n^3+13n^2-4n-3}\right]=\frac{4}{6G+3}$ | 53150 | 4831.8 |
| New | $\mathcal{C}\left[\frac{-2n^4-10n^3-8n^2}{9n^4+48n^3+61n^2-8n-15}\right]=\frac{16}{30G-5}$ | 53147 | 3321.6 |
| New | $\mathcal{C}\left[\frac{-2n^4-12n^3-16n^2}{9n^4+48n^3+61n^2-8n-15}\right]=\frac{64}{90G+65}$ | 53146 | 2214.4 |
| New | $\mathcal{C}\left[\frac{-2n^4-8n^3}{9n^4+48n^3+61n^2-8n-15}\right]=\frac{24}{90G-55}$ | 53145 | 2530.7 |
| New | $\mathcal{C}\left[\frac{-2n^4-10n^3-16n^2-8n}{9n^4+48n^3+85n^2+56n+9}\right]=\frac{8}{27-18G}$ | 53148 | 3126.3 |
| Known | $\phi^2-\phi-1=0$ | 53148 | 8858 |
| Known | $\sqrt{2}^2=2$ | 53147 | 10629.4 |
| Known | $\sqrt{3}^2=3$ | 53147 | 10629.4 |
| Known | $\mathcal{C}\left[\frac{-1}{5}\right]=\frac{\phi+2}{5}$ | 53149 | 5905.4 |
| Known | $\mathcal{C}\left[\frac{-1}{9}\right]=\frac{\phi+1}{3}$ | 53149 | 7592.7 |
| Known | $\mathcal{C}\left[\frac{1}{5}\right]=\frac{\phi+1}{2\phi-1}$ | 53149 | 5905.4 |
| Known | $\mathcal{C}[1]=\frac{\phi+1}{\phi}$ | 53150 | 8858.3 |
| Known | $\mathcal{C}\left[\frac{1}{2}\right]=\frac{\sqrt{3}+1}{2}$ | 53149 | 7592.7 |
| Known | $\mathcal{C}\left[\frac{-1}{8}\right]=\frac{\sqrt{2}+1}{2\sqrt{2}}$ | 53149 | 7592.7 |
| Known | $\mathcal{C}\left[\frac{1}{4}\right]=\frac{1}{2\sqrt{2}-2}$ | 53147 | 6643.3 |
| Known | $9\mathcal{C}\left[\frac{2}{9}\right]^2-9\mathcal{C}\left[\frac{2}{9}\right]-2=0$ | 53153 | 4088.6 |
| New | $\mathcal{C}\left[\frac{6n^2+3n}{n^2+3n+2}\right]=\frac{\phi\sqrt{3}+3\phi+\sqrt{3}}{2\sqrt{3}+2}$ | 53147 | 4088.2 |

| | | | |
|---|---|---|---|
| New | $\mathcal{C}\left[\frac{4n}{2n+3}\right] = \frac{\sqrt{2}\sqrt{3}-2\sqrt{2}-2\sqrt{3}+2}{3-3\sqrt{2}}$ | 53146 | 3126.2 |
| New | $25\mathcal{C}\left[\frac{n^2+3n}{3n^2+9n+6}\right]^2 - 27\mathcal{C}\left[\frac{n^2+3n}{3n^2+9n+6}\right] - 3 = 0$ | 53151 | 3543.4 |
| New | $9\mathcal{C}\left[\frac{4n}{2n+3}\right]^2 - 12\mathcal{C}\left[\frac{4n}{2n+3}\right] - 2 = 0$ | 53152 | 4088.6 |
| Known | $\mathcal{C}\left[\frac{3}{n}\right] = \frac{4C_2e+14e-17}{9}$ | 53145 | 2657.2 |
| Known | $C_2 = \frac{e^2-7}{2}$ | 53146 | 5905.1 |
| New | $\mathcal{C}\left[\frac{2n}{n^2+7n+12}\right] = \frac{1}{15C_2-2}$ | 53147 | 5314.7 |
| New | $\mathcal{C}\left[\frac{-2n}{n^2+9n+20}\right] = \frac{4C_2+14}{5C_2+15}$ | 53152 | 2952.8 |
| New | $\mathcal{C}\left[\frac{-2n}{n^2+7n+12}\right] = \frac{2C_2+7}{2C_2+8}$ | 53151 | 3543.4 |
| New | $\mathcal{C}\left[\frac{2}{n}\right] = C_2 + 2$ | 53151 | 7593 |
| New | $\mathcal{C}\left[\frac{2n+4}{n^2+n}\right] = \frac{2C_2+8}{C_2+3}$ | 53152 | 4088.6 |
| New | $\mathcal{C}\left[\frac{2n}{n^2+3n+2}\right] = \frac{2}{3C_2+1}$ | 53149 | 6643.6 |
| New | $\mathcal{C}\left[\frac{-2n}{n^2+5n+6}\right] = \frac{2C_2+7}{3C_2+9}$ | 53153 | 3543.5 |
| New | $\mathcal{C}\left[\frac{2n}{n^2+5n+6}\right] = \frac{2}{9C_2}$ | 53148 | 6643.5 |
| New | $(36e-81)\mathcal{C}\left[\frac{2n}{4n^2+8n+3}\right]^2 - 18\mathcal{C}\left[\frac{2n}{4n^2+8n+3}\right] - 1 = 0$ | 53143 | 2310.5 |
| New | $(e-1)\mathcal{C}\left[\frac{2n}{4n^2-1}\right]^2 - 2\mathcal{C}\left[\frac{2n}{4n^2-1}\right] - 1 = 0$ | 53147 | 5905.2 |
| New | $(e-1)9\mathcal{C}\left[\frac{-2n}{4n^2+8n+3}\right]^2 - 6e\mathcal{C}\left[\frac{-2n}{4n^2+8n+3}\right] + e = 0$ | 53147 | 3321.6 |
| Known | $\mathcal{C}\left[\frac{-4n^2-2n+2}{16n^4+48n^3+36n^2-5}\right] = \frac{2e-2}{5e-10}$ | 53147 | 3543.1 |
| Known | $\mathcal{C}\left[\frac{1}{16n^2-4}\right] = \frac{e+1}{2e-2}$ | 53148 | 5314.8 |
| Known | $\mathcal{C}\left[\frac{1}{n}\right] = e - 1$ | 53149 | 8858.1 |
| Known | $\mathcal{C}\left[\frac{-n}{n^2+5n+6}\right] = \frac{e}{3}$ | 53149 | 10629.8 |
| Known | $\mathcal{C}\left[\frac{-4n^2-6n}{16n^4+80n^3+132n^2+80n+11}\right] = \frac{3}{33-11e}$ | 53145 | 3543 |
| Known | $\mathcal{C}\left[\frac{-n}{n^2+3n+2}\right] = \frac{e}{2e-2}$ | 53149 | 6643.6 |
| Known | $\mathcal{C}\left[\frac{-n^3-2n^2}{n^4+4n^3+7n^2+6n+3}\right] = \frac{4e}{6e-3}$ | 53148 | 4831.6 |
| Known | $\mathcal{C}\left[\frac{n}{n^2+5n+6}\right] = \frac{1}{18e-48}$ | 53144 | 3542.9 |
| Known | $\mathcal{C}\left[\frac{-n}{n^2+7n+12}\right] = \frac{e}{4e-8}$ | 53147 | 4831.5 |
| Known | $\mathcal{C}\left[\frac{-n}{n^2+11n+30}\right] = \frac{e}{54e-144}$ | 53143 | 2952.3 |
| Known | $\mathcal{C}\left[\frac{n}{n^2+3n+2}\right] = \frac{1}{4e-10}$ | 53147 | 4831.5 |
| Known | $\mathcal{C}\left[\frac{-n}{n^2+9n+20}\right] = \frac{e}{30-10e}$ | 53145 | 4088 |
| Known | $\mathcal{C}\left[\frac{-n^3-3n^2}{n^4+10n^3+33n^2+40n+14}\right] = \frac{6e}{14e-21}$ | 53146 | 3543 |
| New | $\mathcal{C}\left[\frac{n}{n^2+9n+20}\right] = \frac{1}{600e-1630}$ | 53139 | 2125.5 |
| New | $\mathcal{C}\left[\frac{n}{n^2+7n+12}\right] = \frac{1}{96e-260}$ | 53142 | 2657.1 |
| Known | $\mathcal{C}\left[\frac{-2n^2-n+1}{9n^2+15n+4}\right] = \frac{\pi}{16-4\pi}$ | 53150 | 4429.1 |
| Known | $\mathcal{C}\left[\frac{n^2+2n}{4n^2+8n+3}\right] = \frac{4}{3\pi-6}$ | 53150 | 4831.8 |
| Known | $\mathcal{C}\left[\frac{n^2}{4n^2-1}\right] = \frac{4}{\pi}$ | 53151 | 8858.5 |

| | | | |
|---|---|---|---|
| Known | $\mathcal{C}\left[\frac{-2n^2-n}{9n^2+15n+4}\right]=\frac{1}{20-6\pi}$ | 53149 | 4429 |
| Known | $\mathcal{C}\left[\frac{n^2+4n}{4n^2+16n+15}\right]=\frac{8}{15\pi-40}$ | 53147 | 3126.2 |
| Known | $\mathcal{C}\left[\frac{-2n^2-3n}{9n^2+21n+10}\right]=\frac{6}{15\pi-40}$ | 53148 | 3321.7 |
| Known | $\mathcal{C}\left[\frac{1-2n}{9n+9}\right]=\frac{4}{9\pi-24}$ | 53148 | 3543.2 |
| Known | $\mathcal{C}\left[\frac{-2n^2+n}{9n^2+3n-2}\right]=\frac{1}{\pi-2}$ | 53152 | 7593.1 |
| Known | $\mathcal{C}\left[\frac{1-2n}{9n}\right]=\frac{\pi+2}{6}$ | 53153 | 5905.8 |
| Known | $\mathcal{C}\left[\frac{-2n^2-3n+2}{9n^2+33n+28}\right]=\frac{6\pi-32}{448-147\pi}$ | 53145 | 1771.5 |
| Known | $\mathcal{C}\left[\frac{-2n^2-n+1}{9n^2+21n+10}\right]=\frac{4-3\pi}{30\pi-100}$ | 53147 | 2530.8 |
| Known | $\mathcal{C}\left[\frac{-2n^2-3n+2}{9n^2+21n+10}\right]=\frac{2\pi+8}{5\pi}$ | 53151 | 4429.2 |
| Known | $\mathcal{C}\left[\frac{-2n^2+n}{9n^2-3n-2}\right]=\frac{2}{\pi}$ | 53152 | 10630.4 |
| New | $\mathcal{C}\left[\frac{-2n^4-9n^3-9n^2+n+3}{9n^4+36n^3+39n^2+6n-5}\right]=\frac{6\pi+21}{10\pi+25}$ | 53148 | 2530.8 |
| New | $\mathcal{C}\left[\frac{-2n^2-5n+3}{9n^2+39n+40}\right]=\frac{9\pi}{256-72\pi}$ | 53146 | 2310.6 |
| New | $\mathcal{C}\left[\frac{1-2n}{9n+18}\right]=\frac{32-15\pi}{270\pi-864}$ | 53144 | 1610.4 |
| New | $\mathcal{C}\left[\frac{-2n^2-5n+3}{9n^2+33n+28}\right]=\frac{9\pi+24}{56}$ | 53148 | 2952.6 |
| New | $\mathcal{C}\left[\frac{-2n^2+n}{9n^2+21n+10}\right]=\frac{48}{525\pi-1600}$ | 53143 | 1771.4 |
| New | $\mathcal{C}\left[\frac{-2n^2-5n+3}{9n^2+21n+10}\right]=\frac{27\pi+84}{30\pi+100}$ | 53147 | 1898.1 |
| New | $\mathcal{C}\left[\frac{-2n^2-n}{9n^2+21n+10}\right]=\frac{4}{240-75\pi}$ | 53147 | 2530.8 |
| New | $\mathcal{C}\left[\frac{-2n^2-5n+3}{9n^2+27n+18}\right]=\frac{5\pi+16}{6\pi+16}$ | 53150 | 2657.5 |
| New | $\mathcal{C}\left[\frac{-2n^2+n}{9n^2+15n+4}\right]=\frac{3}{15\pi-44}$ | 53149 | 3543.2 |
| Known | $\frac{\pi e}{2}=\left(\frac{1}{\mathcal{C}[n]}+\frac{1}{2\mathcal{C}\left[\frac{1-2n}{4n^2+4n}\right]-1}\right)^2$ | 2946 | 84.1 |
| New | $\frac{\pi e}{2}=\left(\frac{1}{\mathcal{C}[n]}+\mathcal{C}\left[\frac{1}{2n}\right]\right)^2$ | 2947 | 245.5 |
| New | $\frac{\pi e}{2}=\left(\frac{1}{\mathcal{C}[n]}-\frac{\mathcal{C}\left[\frac{n+2}{2n^2+2n}\right]-1}{\mathcal{C}\left[\frac{n+2}{2n^2+2n}\right]-2}\right)^2$ | 2949 | 62.7 |
| Known | $\mathcal{C}\left[\frac{1}{n^2+n}\right]=\frac{1}{C_1}$ | 53151 | 13287.75 |
| New | $\mathcal{C}\left[\frac{4n^2+4n-3}{4n^4+20n^3+39n^2+35n+14}\right]=\frac{3C_1+3}{7C_1}$ | 53149 | 5314.9 |