# OpenReview forum: "The Ramanujan Library - Automated Discovery on the Hypergraph of Integer Relations"
_ICLR.cc/2025/Conference — ICLR 2025 Poster_

### Official Review · Reviewer_AmTb · 2024-11-03

**Soundness:** 4
**Presentation:** 3
**Contribution:** 4
**Rating:** 8
**Confidence:** 3

**Summary:**

The paper presents a new library for automatically discovering functional relations between mathematical constants. The paper outlines the structure of the library as well as the numerical approaches to discovering (polynomial) functional dependencies between fundamental mathematical constants. The authors discuss the convergence properties of the selected approach as well as some limitations and future directions.

**Strengths:**

I am not an expert in the area but I enjoyed reading the paper. There are quite a few things I like about it:

- Fundamental mathematical constants are fascinating, they are often the cornerstones in many scientific disciplines. Discovering new complex relations between them can inspire interdisciplinary discoveries.
- The presented approach for the representation of organizing the relations in a hypergraph and searching for new relations using numerical methods is innovative.
- The implementation is publicly available and likely to benefit the broader scientific community.
- The most time-consuming algorithm is embarrassingly parallel, so it is likely scalable.

**Weaknesses:**

I think the presentation can be improved. The paper is interesting but at times it felt more like reading an article from Quanta magazine. Here are a few suggestions:
- Include a more thorough description of the setup. How was the hypergraph initially created, how many constants and relations were used, how much did the graph expand when new relations were discovered, etc? Some tables and figures can help.
- Include some explanation about operating the library. For example, how can one include new constants and start searching for relations? I think some high-level pseudocode will help.
- Add some formal results about the validity of results and rate of convergence. I think the discussion in section 3 can be organized in a more formal way using some lemmas. This will help the reader concentrate on the results and maybe think about improvements.
- Maybe provide examples with a few more constants, I see only $\pi$, $e$ and $\zeta$.

**Questions:**

Would it be feasible and useful to create a knowledge graph that contains information about the constants and the relations between them? Initially, it will contain human/LLM written explanations, e.g. a short text with references to research papers that introduce the constant or relation and how it is used. When new relations are discovered, the graph will be automatically updated with a short description of the discovery process including computational statistics. The edges of the graph can be labeled with the different relation types (e.g. linear, polynomial, etc.) and/or the certainty of the discovered connection. Knowledge graphs are very popular and this would allow the usage of third-party software for a better visualization and understanding of the discovery process.

---

> ### Author Response · Authors · 2024-11-21
> **Response part 1 to Reviewer AmTb**
>
> **I think the presentation can be improved. The paper is interesting but at times it felt more like reading an article from Quanta magazine. Here are a few suggestions:**
> * **Include a more thorough description of the setup. How was the hypergraph initially created, how many constants and relations were used, how much did the graph expand when new relations were discovered, etc? Some tables and figures can help.**
>
> We thank the referee for the constructive questions. We updated the text to address these questions:
> Section 2.1 now explains the algorithm, starting from a totally disconnected hypergraph of integer relations, and later showing how the algorithm can also accept a partially filled-in hypergraph, using its existing relations to save time.
> Section 5 now specifies that we obtained our results after starting from a totally disconnected hypergraph, containing the constants of our interest. That is, our setup had no relations when initializing the hypergraph. As a consequence of this, the hypergraph was expanded by 118 edges, each corresponding to a relation, all of which are documented in the paper.
> The hypergraph figure appears at the end of the text, after being discussed in Section 5. In its presentation, we omitted "redundant" relations for clarity. Additional tables in the main text and appendices summarize the relations (see tables 2 and 3 for a few examples, and appendix F for a full listing).
>
> * **Include some explanation about operating the library. For example, how can one include new constants and start searching for relations? I think some high-level pseudocode will help.**
>
> We now provide a detailed README providing instructions on how to install Python and operate our library, along with some simple code snippets for retrieving information about the hypergraph of integer relations. The supplementary material also contains the code we used for searching relations on scale, and instructions on how to find relations on one set of constants through our `identify` feature. Both of these are described in the README.
>
> * **Add some formal results about the validity of results and rate of convergence. I think the discussion in section 3 can be organized in a more formal way using some lemmas. This will help the reader concentrate on the results and maybe think about improvements.**
>
> Regarding the validity of results, our work relies on our Return on Investment (RoI) metric to quantify how unlikely a conjecture is to be false. We justify the use of RoI through experiments that show statistical evidence for how RoI may be used in practice. The updated text now better reflects this in section 6.
>
> As an additional formal contribution that strengthens the validity of our results, we also provide 24 formulas that we succeeded in proving, presented in appendix E. We focused there on cases of especially slow convergence, for which the numerical precision was limited to only 10-20 digits. These proofs help validate our approach and show that the results can be relied on in future research efforts. We are working in parallel with mathematicians on general mathematical approaches for proofs that can be applied on scale for the large number of newly discovered formulas.
>
> Regarding the rate of convergence, we followed the suggestion by the referee and now present it as a formal conjecture instead of as a table. We also highlight that certain special cases of this conjecture are proven (see Ben David et al., 2024). Interestingly, the conjecture we pose here is more powerful than the previous results and will hopefully attract mathematicians to pursue the required proof. We agree that this way of placing the results can help organize our findings and clarify what future efforts are required. We thank the referee for this suggestion.
>
> * **Maybe provide examples with a few more constants, I see only pi, e and zeta.**
>
> We replaced one of the results involving $e$ in table 3 with a result involving Catalan’s constant $G$, and updated the text in section 5 to more explicitly mention $ln2$ and the Lemniscate constants that it presents.

---

> ### Author Response · Authors · 2024-11-21
> **Response part 2 to Reviewer AmTb**
>
> **Questions:**
>
> **Would it be feasible and useful to create a knowledge graph that contains information about the constants and the relations between them? Initially, it will contain human/LLM written explanations, e.g. a short text with references to research papers that introduce the constant or relation and how it is used. When new relations are discovered, the graph will be automatically updated with a short description of the discovery process including computational statistics. The edges of the graph can be labeled with the different relation types (e.g. linear, polynomial, etc.) and/or the certainty of the discovered connection. Knowledge graphs are very popular and this would allow the usage of third-party software for a better visualization and understanding of the discovery process.**
>
> We thank the referee for this insightful question. Presenting our hypergraph of relations as a knowledge hypergraph may be a superior presentation for researchers in knowledge representation, and augmenting it with written explanations and relevant papers is a long-term goal of the Ramanujan Library. We have updated the text in section 6 to add discussion of the hypergraph as a knowledge hypergraph.

---

> > ### Comment · Reviewer_AmTb · 2024-11-24
> >
> > I thank the authors for their responses and I maintain my score. Good luck with the paper!

---

### Official Review · Reviewer_Bz6b · 2024-11-03

**Soundness:** 3
**Presentation:** 2
**Contribution:** 2
**Rating:** 3
**Confidence:** 4

**Summary:**

This proposes an improved method to discover equations

**Strengths:**

Automated equation discovery is interesting and providing a library with basic tools is certainly useful.
The text uses good english.

**Weaknesses:**

The text only provides a high level explanation of a few ingredients of the proposed method.  While it may be understandable for a few specialists, the text isnt accessible to the general logic or reasoning expert.
The text doesnt introduce the basics of the considered formalism, does not offer specifications nor examples of inputs or outputs of the system.



#### Supplementary material

The code contains a Readme file but it does not comprehensively explain how to install,or use the code.  Despite looking around to the several files i didnt succeed to use it nor understand it.

#### details

* when defining hypergraph, make clear whether you consider directed / ordered or undirected edges.
* Sec 2 mentions giving cnstants such as $\pi$ or $e$ explicitly but does not say how to give an irrational number explicitly.
* please specify how one can define a C teansform (as it contains an infinite number of parameters)
* while libe 108 talks about polynomial relations, line 111 only gives the form of a linear relation
* line 115 requires thar the absolute value of the linear expression equals exactly $\epsilon$.  I suppose you mean $\le$ instead.
* the precision is defined as $\log(\epsilon)$, i guess this can lead to problems if $\epsilon$ happens to be exactly 0 ?
* line 119: "we define its degree as the sum of all exponents in each monomial": does this imply you consider homogeneous polynomials, as else the different monomials may have a different sum of exponents?
* line 125: is an edge a hyperedge?
* line 127: please define transitivity in this context
* line 153: please define "type"
* line 155: the expression "product space on certain subtypes" is very unclear.
* line 159: it is unclear what is a combined constant, or what language or equations the user can use to define them.

Section 5 may have provided useful insight in capabilities of the proposed system,if the used notations would have been clear.

Making abstraction of that problem i observe the displayed equations are rather complicated and hence the search space of possible conjectures must be huge.  Given one searches only approximate equalities under bounded precision it seems likely one will discover many incorrect conjectures.

**Questions:**

It may help to answer questions in my detailed comments, even if that is not guaranteed to clarify everything.

---

> ### Author Response · Authors · 2024-11-21
> **Response part 1 to Reviewer Bz6b**
>
> **The text only provides a high level explanation of a few ingredients of the proposed method. While it may be understandable for a few specialists, the text isnt accessible to the general logic or reasoning expert. The text doesnt introduce the basics of the considered formalism, does not offer specifications nor examples of inputs or outputs of the system.**
>
> We thank the referee for this comment. We improved the manuscript in two aspects: (1) We have rewritten the text so it is more accessible, and better presents the reasoning. (2) We provide concrete basic examples of inputs and outputs of the system, both at the level of populating the hypergraph, and at the level of automated discovery of a single edge.
>
> (1) To make the text more accessible, we revised the introduction so it now contains a paragraph providing a high-level explanation of our work and its contribution, explaining the general logic. To summarize this logic here, mathematical constants are normally associated with unrelated scientific disciplines. However, it is possible to relate them through mathematical formulas, leading to surprising connections. One such connection is the solution to the Basel problem, namely $\zeta(2)=\pi^2/6$. Another is the following formula by Ramanujan, featuring an unusual combination of $\pi$ and $e$:
> $$\sqrt{\frac{\pi e}{2}}=\cfrac{1}{1+\cfrac{1}{1+\cfrac{2}{\ddots+\cfrac{n}{1+\ddots}}}} +
> 1+\frac{1}{3}+\frac{1}{15}+\cdots+\frac{1}{(2n-1)!!}$$
> One contribution of our work is to automatically discover and catalogue such relations between constants, resulting in a hypergraph of integer relations. As more relations are added to this hypergraph, it better captures the full network of formulas that connect the constants.
>
> (2) To provide concrete examples for inputs and outputs of the system, we updated section 2.1 so it first explains the algorithm as it operates from a totally disconnected hypergraph of integer relations, and later shows how the algorithm can also accept a partially filled-in hypergraph, using its existing relations to save time.
>
> In addition, the caption of table 3 has been extended with an example, showing how the relation $C\left[\frac{-2n^4}{9n^4 - 3n^2 + 1}\right] = \frac{1}{2G}$ can be found using PSLQ:
>
> _“Assuming a polynomial with degree $2$ and order $1$, and given Catalan's constant $G$ and $C\left[\frac{-2n^4}{9n^4 - 3n^2 + 1}\right]$, PSLQ can find integers $n_1,n_2,n_3,n_4$ such that $n_1C\left[\frac{-2n^4}{9n^4 - 3n^2 + 1}\right]G + n_2C\left[\frac{-2n^4}{9n^4 - 3n^2 + 1}\right] + n_3G + n_4 = 0$. In this case, one can find $n_1=2, n_2=0, n_3=0, n_4=-1$, and given that both $C\left[\frac{-2n^4}{9n^4 - 3n^2 + 1}\right]$ and $G$ are known to high precision, this is a signal that an integer relation exists. The other relations here can be captured in a similar way, using more complex integer polynomials."_
>
> **Supplementary material**
>
> **The code contains a Readme file but it does not comprehensively explain how to install,or use the code. Despite looking around to the several files i didnt succeed to use it nor understand it.**
>
> We thank the referee for expressing interest in the supplementary material. We have improved the README contained within to give more detailed instructions for installing Python and using the code.
>
> **details**
> * **when defining hypergraph, make clear whether you consider directed / ordered or undirected edges.**
>
> We now clarify that our hypergraph is undirected, when it is first introduced in section 2.
>
> * **Sec 2 mentions giving cnstants such as pi or e explicitly but does not say how to give an irrational number explicitly.**
>
> We thank the referee for this comment. In theory, one may use irrational numbers symbolically, but in practice one provides the digits of any such number up to a chosen precision. This is now clarified in the text.
>
> * **please specify how one can define a C teansform (as it contains an infinite number of parameters)**
>
> We thank the referee for this comment. We now clarify in the text that a C transform is defined using an arbitrary complex sequence $f_n$. In practice, $f_n$ will be generated from a rational function, which means that the space of all such C transforms is indeed (countably) infinite.
>
> * **while libe 108 talks about polynomial relations, line 111 only gives the form of a linear relation**
>
> We thank the referee for this observation. The text has been clarified so it introduces the concept of polynomial relations gradually, through integer relations.
>
> * **line 115 requires thar the absolute value of the linear expression equals exactly epsilon. I suppose you mean <= instead.**
>
> We apologize for the confusion we may have caused in our wording. We in fact define and use $\varepsilon$ as the exact numerical error, so it is not an inequality. The text is now rewritten to reflect this:
> $\varepsilon:=|a_1x_1+a_2x_2+...|\geq 0$

---

> ### Author Response · Authors · 2024-11-21
> **Response part 2 to Reviewer Bz6b**
>
> * **the precision is defined as log(epsilon), i guess this can lead to problems if epsilon happens to be exactly 0 ?**
>
> We thank the referee for this deduction. In theory, $\varepsilon=0$ implies a true integer relation with no inaccuracy, whose precision is $+\infty$. However, in practice, we may replace $\varepsilon$ with the numerical inaccuracy of the least precise constant involved in the relation, thus avoiding this issue. This is now clarified in the text.
>
> * **line 119: "we define its degree as the sum of all exponents in each monomial": does this imply you consider homogeneous polynomials, as else the different monomials may have a different sum of exponents?**
>
> We thank the referee for spotting this mistake. The text is now amended to specify the *greatest* sum of all exponents in each monomial.
>
> * **line 125: is an edge a hyperedge?**
>
> We apologize for the confusion. Yes, an edge is a special case of a hyperedge, but can also be used interchangeably, as we have in the text. In the updated paper, we now primarily use the term edge, simplifying the text.
>
> * **line 127: please define transitivity in this context**
>
> We now define transitivity in the place where it is mentioned.
>
> * **line 153: please define "type"**
>
> We apologize for the confusion the word may have caused in this situation. The text has been rewritten to refer to arbitrary partitioning of the space of constants, without using this word.
>
> * **line 155: the expression "product space on certain subtypes" is very unclear.**
>
> We thank the referee for this comment. The text has been rewritten to be clearer about how the product space is created.
>
> * **line 159: it is unclear what is a combined constant, or what language or equations the user can use to define them.**
>
> We apologize for the confusion we may have caused here. Truthfully, $\pi e$ is as valid a choice of constant as any other. We removed the word *combined* from the relevant sentence in the text.
>
>
> Altogether, we were glad to revise the manuscript in all the aspects suggested by the referee. These improvements helped clarify it and we are grateful for these suggestions.
>
> **Section 5 may have provided useful insight in capabilities of the proposed system,if the used notations would have been clear.**
>
> Section 5 makes use of notation we have defined in section 2. The text is now clarified with a reference to the relevant definition.
>
> **Making abstraction of that problem i observe the displayed equations are rather complicated and hence the search space of possible conjectures must be huge. Given one searches only approximate equalities under bounded precision it seems likely one will discover many incorrect conjectures.**
>
> We are thankful for the referee expressing concern regarding this issue. The likelihood of a conjecture being incorrect is something we tackle in our paper using a metric we call Return on Investment (RoI). This approach allowed us to identify conjectures that are extremely unlikely to be incorrect. These are the conjectures presented in our manuscript. The RoI metric expresses how unlikely it is for a conjecture of a certain precision to be obtained given the number of integer digits used. This metric is an essential part of our algorithm, as it is how conjectures that are likely to be true are automatically distinguished from noise.
> Section 5 in the updated text now references the relevant section 3, where the exact definition of RoI, along with experiments empirically showing its validity, can be found.
>
> **Questions:**
> **It may help to answer questions in my detailed comments, even if that is not guaranteed to clarify everything.**
>
> We were glad to answer these questions and revise the manuscript accordingly. If there are any other points to clarify, please let us know.

---

> > ### Comment · Reviewer_Bz6b · 2024-11-24
> >
> > Thanks for your detailed answer.
> > Your answer makes clearer how your algorithm works, and allows me to start thinking about its implications.
> > While the revisions clearly improve the text, I'm still concerned about the quality of the explanation (e.g., it seems still unclear to the reader what the user can exact do or not do with the algorithm/software, e.g., what is possible input, what parameters can be selected, what are considerations to set parameters ...) and the lack of analysis (e.g., I understand that RoI allows somehow to assess how surprising a relation is depending on the description length of the discovered equation and its precision, but it is unclear whether there is proven theory supporting this notion and what assumptions are made, e.g., it is also important to consider the number of conjectures considered as considering more potential equations increases the probability of incorrectly assigning an equation a too low RoI.  Also, there isn't much information on practical runtime and problem sizes, next to a few asymptotic upper bounds as in appendix C).

---

> ### Author Response · Authors · 2024-11-28
>
> **Thanks for your detailed answer. Your answer makes clearer how your algorithm works, and allows me to start thinking about its implications.**
>
> We thank the referee for considering our answer, and are glad to know that it helped with the understanding of our work.
>
> **While the revisions clearly improve the text, I'm still concerned about the quality of the explanation (e.g., it seems still unclear to the reader what the user can exact do or not do with the algorithm/software, e.g., what is possible input, what parameters can be selected, what are considerations to set parameters ...)**
>
> We thank the referee for reviewing the updated manuscript and for raising this concern. The Ramanujan Library provides an API with several functionalities:
> * Drawing information from the database, including constants and integer relations. Examples include:
>   * `db.constants` retrieves all “famous” constants.
>   * `db.relations()` retrieves all integer relations.
> * The `identify` function, used for numerical identification tasks.
> * Contributing new relations to the database is done primarily through running our search algorithm, which directly uploads its results to the database.
>
> These capabilities are now explained in the README found in the supplementary material.
>
> **[,,,] lack of analysis (e.g., I understand that RoI allows somehow to assess how surprising a relation is depending on the description length of the discovered equation and its precision, but it is unclear whether there is proven theory supporting this notion and what assumptions are made**
>
> We would like to take this opportunity to provide a demonstration of Return on Investment (RoI) and how it can be used to detect and/or reject integer relations in practice:
>
> Consider the constant $C\left[\frac{-n^6}{4n^6 + 483n^4 + 14763n^2 - 3721}\right] \approx 0.9999131740…$. In appendix F we list this as part of a conjectured relation connecting it to $\zeta(3)$. Depending on the precision used, PSLQ will provide different conjectures:
>
> Using 60 binary digits, PSLQ will provide the conjecture $C\left[\frac{-n^6}{4n^6 + 483n^4 + 14763n^2 - 3721}\right] = \frac{114491\zeta(3)-30033}{36579\zeta(3)+63631}$. Regardless of whether this conjecture is true, RoI immediately allows one to dismiss it as likely being noise, since the RoI evaluates to approximately $0.67$.
>
> Increasing the binary precision to 110, PSLQ provides the same conjecture we claim in the paper (in appendix F, row 15 in the table):
>
> $$C\left[\frac{-n^6}{4n^6 + 483n^4 + 14763n^2 - 3721}\right] = \frac{216000}{13176000\zeta(3) - 15622283}$$
>
> However, at this point it is not yet notable, as the RoI is still $1.15$.
>
> Finally, increasing the binary precision to 445, PSLQ still provides the same conjecture, but with the increased precision it is now a much more viable conjecture, providing an RoI of $5.9$ as we state in the paper. At this point, there is high enough confidence that PSLQ will not change the conjecture it finds even if one were to increase the precision further, implying that this conjecture is likely to be true.
>
> This example points to a general approach for getting arbitrarily high confidence in the correctness of a conjectured formula: one can go on increasing the binary precision of PSLQ, and show that the same formula keeps emerging with higher and higher RoI measures.
> Note that this demonstration can be thought of as a special case of the large-scale experiments that we show in figures 3 and 7. Under this lens, one can interpret the gradual increase of binary precision as gradually denoising the integer relation, pushing it further away from results typical of random noise.
>
> **[...] considering more potential equations increases the probability of incorrectly assigning an equation a too low RoI.**
>
> Return on Investment is determined exactly by the accurate digits involved in an integer relation. RoI may appear low when one constant is significantly less accurate than the other constants involved. This can happen when such a constant is provided by a formula with slow convergence. This is an unavoidable problem that requires either finding formulas with significantly better convergence, or brute force computation of the existing formula. Fortunately, RoI is easy to recalculate once more digits are available.
>
> **[...] there isn't much information on practical runtime and problem sizes**
>
> We thank the reviewer for this suggestion. We clarified the text in section 5 to mention that the total compute hours that we ran the algorithm to obtain our results is about 16 compute months, while mentioning the embarrassingly parallel nature of our algorithm, which means the runtime in practice goes down with the number of compute cores used. This means that the practical runtime is as low as one has access to large amounts of CPU cores.

---

### Official Review · Reviewer_cctr · 2024-11-05

**Soundness:** 4
**Presentation:** 3
**Contribution:** 2
**Rating:** 8
**Confidence:** 3

**Summary:**

The paper introduce a new search algorithm for integer polynomial relations between mathematical constants. The authors organize their algorithm into a library and use a hypergraph as data structure to store the relations. The library can interact with a relation finding machine to search for new relations.

**Strengths:**

Strengths:
- The paper claims to have discovered new integer relations between constants that have not been discovered before. This is an exciting and relevant direction for publication in ICLR since machine assisted mathematics has many unrevealed potentials.
- The methodology is sound and correct, since it is a combination of previously published and peer-reviewed systems.
- The ROI scheme, while simple, may prove to be generalizable to other domains/applications.
- The resulting code seems to be available publicly.
- The paper is mostly written in a clear and comprehensive way, with appropriate figures and tables for demonstration.

**Weaknesses:**

Some weaknesses include:
- There is almost no machine learning (emphasis on the 'learning' part) in the paper, as it seems to be an engineering/database product that interfaces with existing code base. This is, of course, not a weakness on the content of the paper, but on its fit for the venue of publication.
- The use of hypergraphs as data structure, while theoretically clean, is perhaps only practically useful if the code can make use of graph symmetries (for instance in a graph learning framework). Due to the lack of practical ways to distinguish sets vs sequence in hardware, I very much doubt storing relations as hypergraphs is much more practically advantageous than any naive data structure. Of course, some data structure is needed to store the relations, but the authors also claimed that this is an "effective representation" (line 72) and overall feature it as one of their main contributions.
- From my understanding, perhaps inherent to the problem the authors are trying to solve, until a proof is given, it is almost never guaranteed that any new relations found are actually correct. This makes the problem rather ill-defined since there are no performance metric attributed to the proposed methodology, making it hard, almost impossible, to judge its effectiveness, outside of engineering perspective.

Overall, this is still an interesting paper, even as an engineering paper so I still advocate for publication just for the code and methodology to be published for more to see.

**Questions:**

Some other minor comments and questions:
- Line 42, I believe it is quite a stretch to claim that any computer assisted proof (which comprises of the majority of citations in this sentence are examples of "usage of AI as a scientific tool"
- Line 101-104, it took me awhile to parse the sentences here. This can be solved by explicitly stating the definition of C-transform and how it differs from continued fraction since at first glance and without definition, it's not easy to distinguish. Alternatively, one can also just give an example of the difference. An example of why this is confusing is: it seems that C-transform, as stated on arbitrary function f_n, captures all continued fraction; yet some continue fractions cannot be converted to infinite sums when C-transforms can?
- What is the question mark in table 2?

---

> ### Author Response · Authors · 2024-11-21
> **Response part 1 to Reviewer cctr**
>
> * __There is almost no machine learning (emphasis on the 'learning' part) in the paper, as it seems to be an engineering/database product that interfaces with existing code base. This is, of course, not a weakness on the content of the paper, but on its fit for the venue of publication.__
>
> We thank the referee for raising this issue. Our work is concerned with automated conjecture generation in number theory, which has so far proven difficult to penetrate with classical machine learning and neural network techniques. This field is uniquely challenging even within mathematics, where automated theorem proving (rather than conjecture generation) has seen much more successful applications of machine learning methods such as reinforcement learning (e.g., Fawzi et al. “Discovering faster matrix multiplication algorithms with reinforcement learning”, Nature 610, 47 (2022)). Still, such methods remained ineffective for conjecture generation in number theory.
>
> In that regard, our work is one of a few pioneering efforts, finding successful algorithms for conjecture generation. Despite not relying on classical machine learning or neural networks, the systematic data-driven research and the use of advanced lattice-based algorithms mark our work as the most advanced large-scale effort ever executed for finding new relations between constants in number theory. As such, it is a pioneering work that can attract attention from the wider ICLR community to this new, untapped domain.
>
> As a historical example, consider the simple alpha-beta pruning used in the Stockfish chess AI. Despite now being outdated, the impact of Stockfish nevertheless marks the beginning in pushing the field of chess engines forward, serving as a benchmark for future engines.
> One way we foresee our work being used is as such a benchmark, supporting the development of hybrid approaches to conjecture generation. The hypergraph of integer relations that we proposed and built can be now fed to a neural network as training data. The relevant data is created here systematically for the first time.
>
> Regarding the emphasis on the “learning” part, we would like to note that as the hypergraph of integer relations is populated, future runs of our algorithm become more efficient, as they know where computation is not needed anymore. This results in a learning process, where the computer learns about the hypergraph of integer relations and uses it when looking to further expand on it. This point has been clarified in the updated manuscript, at the end of section 2.1.
>
> This is why we propose this work for ICLR. We hope that through the motivation provided in this work, the wider audience of AI researchers will help the mathematics community to develop new techniques for automated conjecture generation.
>
> * __The use of hypergraphs as data structure, while theoretically clean, is perhaps only practically useful if the code can make use of graph symmetries (for instance in a graph learning framework). Due to the lack of practical ways to distinguish sets vs sequence in hardware, I very much doubt storing relations as hypergraphs is much more practically advantageous than any naive data structure. Of course, some data structure is needed to store the relations, but the authors also claimed that this is an "effective representation" (line 72) and overall feature it as one of their main contributions.__
>
> We thank the referee for bringing up this point. What makes the hypergraph the most effective representation is the fact that integer relations naturally admit a hypergraph structure, and so we chose to organize them in this way. i.e., the hypergraph is the natural structure that emerges, and thus it is used to store the relations (while it is not itself used in the algorithm).
>
> This presentation lends itself to exploring transitivity properties of the hypergraph. Namely, given two relations, what other relations can be constructed using them? This point is clarified in the updated text, as we now define this notion of transitivity in the updated section 2.

---

> ### Author Response · Authors · 2024-11-21
> **Response part 2 to Reviewer cctr**
>
> * __From my understanding, perhaps inherent to the problem the authors are trying to solve, until a proof is given, it is almost never guaranteed that any new relations found are actually correct. This makes the problem rather ill-defined since there are no performance metric attributed to the proposed methodology, making it hard, almost impossible, to judge its effectiveness, outside of engineering perspective.__
>
> We thank the referee for bringing up this point. Regarding a performance metric, we use a quantifiable method involving the number of digits of accuracy in a discovered conjecture, compared to the amount of integer digits used. This approach provides a measure for predicting the likelihood for the correctness of each discovered formula.
> We quantify this approach using a metric we call Return on Investment (RoI). For example, consider the formulas we show in figure 3, each with RoI in the thousands. Our algorithm assumes a minimal RoI of 2, which is already good enough to separate significant formulas from noise. This is why the problem is no longer ill-defined, but follows a clear performance metric.
> We updated section 2.1 so it references the following section 3, where we discuss RoI in detail and provide experiments that justify its use in practice.
>
> As additional evidence of the reliability of our approach, we also provide 24 formulas that we succeeded in proving, presented in appendix E. We focused there on cases of especially slow convergence, for which the numerical precision was limited to only 10-20 digits. These proofs help validate our approach and show that the results can be relied on in future research efforts. We are working in parallel with mathematicians on general mathematical approaches for proofs that can be applied on scale for the large number of newly discovered formulas.
>
> Let us add more generally about the importance of finding new unproven formulas (or generally conjectures): The value of such discoveries in mathematics cannot be understated, as it is usually such a conjecture that acts as the first step in mathematical research, which eventually leads to a discovery of a new theory. As an example, consider Srinivasa Ramanujan’s contributions to mathematics, many of which were initially conjectures and were not proven by him, yet his impact on the mathematical world is undeniable. In a similar way, the conjecture generation algorithm in our work provides new leads for mathematical research that can have long-term impact.
> Historically, conjecturing often relied on human intuition or creativity, and only recently has it benefited from automation. This is where our contribution lies, within the field of automated conjecture generation.
>
> __Overall, this is still an interesting paper, even as an engineering paper so I still advocate for publication just for the code and methodology to be published for more to see.__
>
> We thank the referee for the kind words. The code is currently available in the supplementary material, and will be replaced with a Github link in the camera-ready version as well. We also added a tutorial to the README in the code, showing queries against the database provided within.
> * __Line 42, I believe it is quite a stretch to claim that any computer assisted proof (which comprises of the majority of citations in this sentence are examples of "usage of AI as a scientific tool"__
>
> We thank the referee for bringing this up. This terminology was used in the literature in the old days, where the definition of AI was broader. We revised this sentence to now say “usage of computer algorithms as scientific tools”.
> * __Line 101-104, it took me awhile to parse the sentences here. This can be solved by explicitly stating the definition of C-transform and how it differs from continued fraction since at first glance and without definition, it's not easy to distinguish. Alternatively, one can also just give an example of the difference. An example of why this is confusing is: it seems that C-transform, as stated on arbitrary function f_n, captures all continued fraction; yet some continue fractions cannot be converted to infinite sums when C-transforms can?__
>
> We apologize for the confusion we have caused with these lines. Continued fractions, in general, generalize infinite sums. Certain continued fractions can be converted to simple infinite sums, but the majority cannot be. C-transforms are used as a canonical representation that can be applied to any continued fraction, in a way that unifies infinitely many continued fractions. Thus C-transforms are also more expressive than infinite sums. The updated text now reflects this, and includes a more explicit definition of C-transforms.
> * __What is the question mark in table 2?__
>
> This question mark denotes a case of a C-transform for which there is currently no formula that predicts its error at a given evaluation depth. In the updated text, the question mark is replaced with an “N/A”, which is mentioned in the caption.

---

> > ### Comment · Reviewer_cctr · 2024-11-27
> >
> > Thank you for your response and for addressing my questions. I am increasing my score to an 8.

---

### Meta-Review · Area_Chair_WFGB · 2024-12-19

**Metareview:**

The main claim of the paper is an approach for learning a hypergraph of relations between integers. This can be useful for people working in number theory to discover new relationships that were not known or known to the researcher. As the reviewers note, this is not a typical paper for this community, but there was significant interest in the ideas presented.

The strengths were that the ROI scheme may prove to be generalizable to other domains/applications. The introduction of new ideas and problems to the ICLR community was viewed as a significant strength.

The weaknesses were that that the paper may not be viewed as significant to those outside the domain/field of number theory. The clarity of the exposition was also a cause of concern for some reviewers.

The effort of the authors to connect this work to the ICLR community. They seem to have made use of the technology the community has developed, adapted it, and improved it for their use. While the reviews for this paper had a wide range, there seems to have been sufficient interest and consensus that the claims rest firmly on the evidence to support acceptance. The concerns about exposition and clarity have been taken seriously by the authors and the paper has benefited greatly from the advice, comments and engagement of the reviewers.

**Additional Comments On Reviewer Discussion:**

A question that arose in the discussion was what the differences are between the hypergraphs in this paper and traditional knowledge graphs. Here the reviewers were very helpful to articulate the differences. The authors are encouraged to describe the relationship between their work and traditional knowledge graphs so that the ICLR community can better place this work into context. The reviewers also helped to contextualize the contributions of the work.

The authors, appropriately, focused on the concerns about clarity and exposition. This seems to be a topic that is outside of the focus of most of the community, but has drawn interest from the reviewers. I expect that others in the community will find similar interest in the paper even if it doesn't immediately have a direct impact on their day-to-day work. There is value in pushing the boundaries in pure mathematics where they come into contact with the core competencies of the ICLR community. I think the reviews indicate the authors have made such an effort.

---

### Decision · Program_Chairs · 2025-01-22

Accept (Poster)